

# The upstream-downstream mechanism of North Atlantic and Mediterranean cyclones in semi-idealized simulations

Alexander Scherrmann[1], Heini Wernli[1], and Emmanouil Flaounas[2]

[1]ETH Zurich, Institute for Atmospheric and Climate Science, Zurich, Switzerland
[2]Hellenic Centre for Marine Research, (HCMR), Anavyssos, Greece

**Correspondence:** Alexander Scherrmann (alexander.scherrmann@env.ethz.ch)

**Abstract.** Cyclogenesis in the Mediterranean typically follows an archetypal scenario with the intrusion of a potential vorticity (PV) streamer over the Mediterranean, which results from a preceding Rossby Wave breaking (RWB) upstream over the North Atlantic. The RWB is amplified by the presence of warm conveyor belts (WCBs) in at least one North Atlantic cyclone about 4 days prior to Mediterranean cyclogenesis. This scenario has been found in several case studies of intense Mediterranean cy-
clones with a variety of different North Atlantic cyclone tracks and intensities, and different locations of PV streamers intruding the Mediterranean. While four main events (namely North Atlantic cyclone, WCBs, RWB and the resulting PV streamers) are suggested to be necessary for Mediterranean cyclogenesis, they rarely occur in a spatially consistent, fully repetitive pattern. To more systematically study this link between the upstream North Atlantic cyclone dynamics and the Mediterranean cyclogenesis, we perform a set of semi-idealized simulations over the Euro-Atlantic domain. For these simulations, we prescribe
a constant climatological atmospheric state in the initial and boundary conditions. To trigger the downstream Mediterranean cyclogenesis scenario, we perturb the climatological polar jet through the inversion of a positive upper-level PV anomaly. The amplitude of this perturbation determines the intensity of the triggered North Atlantic cyclone. This cyclone provokes RWB, the intrusion of a PV streamer over the Mediterranean, and the formation of a Mediterranean cyclone. Therefore, our results show a direct causality between the presence of a North Atlantic cyclone and the downstream intrusion of a PV streamer into
the Mediterranean, which causes cyclogenesis about 4 days after perturbing the polar jet, which we refer to as the upstream-downstream mechanism of cyclogenesis. To investigate the sensitivity of this mechanism, we vary the position and amplitude of the upper-level PV anomaly. In all simulations, cyclogenesis occurs in the Mediterranean. Nevertheless, the evolution, track and intensity of the Mediterranean cyclones are sensitive to the dynamical structure and intensity of the intruding PV streamer, which itself is sensitive to the interaction of the upstream cyclone and the RWB. By applying different seasonal climatological
atmospheric states as initial conditions we show that the seasonal cycle of Mediterranean cyclogenesis indeed depends on the large-scale atmospheric circulation. In particular, we show that the Mediterranean cyclones in these semi-idealized simulations show characteristics that agree with the observed climatology of Mediterranean cyclones in the respective season.



## 1   Introduction

Mediterranean cyclones are frequent and potentially high-impact weather systems in one of the most densely populated regions of the world. In fact, the Mediterranean basin belongs to the regions of highest winter cyclone activity around the globe (Trigo et al., 1999; Ulbrich et al., 2009; Neu et al., 2013). Mediterranean cyclones typically form due to Rossby wave breaking (RWB) in the upper troposphere and the consequent southward intrusion of stratospheric air of high potential vorticity (PV; PV streamers; Appenzeller and Davies, 1992) over the region (Raveh-Rubin and Flaounas, 2017). The spatially distinct regions of

cyclogenesis in the Mediterranean (e.g. Bartholy et al., 2009; Almazroui et al., 2015) suggest that there is a high variability in the spatial structure of the PV streamers that results in Mediterranean cyclogenesis. Indeed, it has been shown by several case studies (e.g. Grams et al., 2011; Chaboureau et al., 2012, Pantillon et al., 2013, Portmann et al., 2020) that, depending on where the PV streamer intrudes into the Mediterranean, cyclogenesis occurs in different parts of the Mediterranean and the resulting cyclones may affect different areas in the basin with different impacts. For instance, PV streamers and the resulting

cyclones in the western parts of the Mediterranean, e.g., the Gulf of Genoa potentially lead to heavy precipitation south of the Alps (Massacand et al., 1998; Argence et al., 2008). But the RWB can also lead to PV streamers further east with cyclones affecting the eastern Mediterranean (e.g. Portmann et al., 2020).

The presence of the PV streamer is regarded as essential to trigger cyclogenesis (Massacand et al., 1998; Emanuel, 2005; Fita et al., 2006; Chaboureau et al., 2012; Flaounas et al., 2015; Raveh-Rubin and Flaounas, 2017), and, in addition, several

studies have shown that the detailed structure of the streamer is essential for determining the cyclone's track and intensity (e.g. Flocas, 2000; Romero, 2001; Homar and Stensrud, 2004; Argence et al., 2008; Claud et al., 2010; Carrió et al., 2017). Despite the importance of the PV streamer, our knowledge on the chain of events that determines the location of its intrusion over the Mediterranean and its detailed structure are still not fully clear.

Indeed, Portmann et al. (2020) highlighted the importance of a credible representation of the upstream RWB for the accurate

forecasting of the intrusion of the PV streamer and the consequent cyclogenesis. They showed that forecast errors of the location of a PV streamer over the Mediterranean were significantly reduced, once the outflow of a warm conveyor belt (WCB; Madonna et al., 2014) of an upstream North Atlantic cyclone was represented accurately in the initial conditions. WCBs can amplify the ridge by transporting low-PV air from the lower to the upper troposphere (condensation inside the ascending airstreams results in PV destruction above) and thereby lead to the RWB downstream of the amplified negative PV anomaly.

Therefore, North Atlantic WCBs seem to play a crucial role in forecasting Mediterranean cyclones. Furthermore, e.g. Pantillon et al. (2013) showed that the North Atlantic cyclone is essential for the RWB and the intrusion of a PV streamer into the Mediterranean, as by removing the North Atlantic cyclone from the initial conditions of a forecast, they could suppress the formation of the downstream Mediterranean cyclone.

Raveh-Rubin and Flaounas (2017) showed that there is a systematic scenario that links Mediterranean cyclogenesis to

different preceding atmospheric features. This scenario includes one or several extratropical cyclones over the North Atlantic. From those cyclones, WCBs originate and amplify the ridge downstream (typically over the central or eastern North Atlantic). The anticyclonic circulation induced by the (amplified) ridge provokes the RWB, which causes the intrusion of a PV streamer





over the Mediterranean. Finally, the cyclonic circulation induced by the upper-level streamer contributes to surface cyclogenesis in the Mediterranean. In summary the chain of events that leads to Mediterranean cyclogenesis includes (i) North Atlantic
cyclone(s), (ii) WCBs amplifying a ridge, (iii) RWB, and (iv) an intruding PV streamer. Raveh-Rubin and Flaounas (2017) identified these features in the ERA-Interim reanalysis as coherent spatial objects and connected them in time. They showed that this scenario occurred in 181 out of the 200 intense Mediterranean cyclones they investigated. They found that 1-5 North Atlantic cyclones are connected to each Mediterranean cyclone downstream, suggesting that there is indeed an upstream-downstream connection between cyclones over the North Atlantic and in the Mediterranean. Although the same preceding
atmospheric features are present for almost all Mediterranean cyclone events, the large-scale atmospheric conditions, i.e., the location of the jet and already acquired wave activity, are highly variable. Therefore, every case of Mediterranean cyclogenesis might be unique even if the features (i-iv) were always present. Hence, it is unknown how the location and intensity of the polar jet and of the North Atlantic cyclone affect the consequent intensity and location of the downstream Mediterranean cyclone.

In these regards, idealized model simulations may serve to better understand this upstream-downstream connection of cy-
clones as they can be operated with reduced complexity, e.g., less internal variability, compared to realistic simulations. Using such an idealized framework, several studies (e.g. Schär and Wernli (1993); Wernli et al. (1999); Schemm et al. (2013)) showed that downstream cyclogenesis can be triggered by locally perturbing the zonal jet, and that moist processes affect the intensity of the downstream cyclone. However, these studies used a so-called channel setup with zonally uniform, analytically prescribed atmospheric conditions, which are not directly comparable to the (sub-)daily and zonally varying atmospheric states over the
North Atlantic. Furthermore, they do not involve the unique geographical features of the Mediterranean basin, which have been shown to be a key factor for cyclone dynamics in the Mediterranean (e.g. Flaounas et al., 2022; Scherrmann et al., 2023).

To contribute towards closing the gap in understanding the connection between the upstream and downstream cyclones, here the North Atlantic and Mediterranean cyclones, respectively, we introduce a semi-idealized modelling framework. In this framework, we trigger cyclogenesis in climatological flow conditions over the North Atlantic and investigate the chain
of events suggested in the literature, i.e., the formation of a cyclone over the North Atlantic, the downstream ridge, RWB, intrusion of the PV streamer, and eventually the cyclogenesis in the Mediterranean. To study the characteristics and sensitivity of this upstream-downstream mechanism, we trigger cyclogenesis over the North Atlantic in a similar way as done in previous baroclinic channel simulations, i.e., by placing a finite-amplitude, localized, positive upper-level PV anomaly with varying amplitude in different locations relative to the upper-level jet. With this approach we address the following objectives:

1. Develop a semi-idealized modelling framework that demonstrates the upstream-downstream cyclogenesis mechanism.

2. Determine the sensitivity of Mediterranean cyclogenesis to the location and intensity of the upstream cyclone over the Atlantic ocean.

3. Investigate the sensitivity of Mediterranean cyclogenesis to the initial large-scale flow, representing climatological mean conditions in different seasons.

The study is structured as follows: In Sect. 2 we provide an overview of the model, the setup of the simulations, and algorithms used, before showing that the upstream-downstream cyclogenesis mechanism occurs in our simulations for winter





in Sect. 3. The results of the sensitivity analysis with varying initial perturbations are presented in Sect. 4. Section 5 provides results where we vary the initial basic state by choosing the climatological mean flow from another season. In the last section we summarize and discuss our results and give an outlook for future studies.

## 2 Numerical experiments and methods

### 2.1 Modelling approach

In this study, we use the Weather, Research and Forecasting Model (WRF; version 4.3; Skamarock et al., 2021). The domain of our experiments covers the North Atlantic-European domain (Fig. 1a) with a horizontal grid spacing of $0.5° \times 0.5°$ and we use 45 vertical hybrid sigma-pressure levels, which is adequate to reproduce Mediterranean cyclogenesis and dynamics (Flaounas et al., 2013). Simulations with a nested domain over the Mediterranean of $0.1°$ resolution provided similar results and conclusions within the variability of cyclone intensities and tracks presented below. The constant in time initial and boundary conditions consist of climatological averages calculated from ERA5 reanalyses (Hersbach et al., 2020) for a particular season. For all simulations, the duration has been set to nine days. In our experiments microphysical processes are reproduced by a one–moment 5–class scheme (Hong et al., 2004) and convection is parameterized by the Kain-Fritsch scheme (Kain, 2004). Both long-wave and short-wave radiation are taken from the RRTMG scheme (Iacono et al., 2008). We use the Yonsei University Scheme (Hong et al., 2006) for the planetary boundary layer parameterization, and for land processes a 5-layer thermal diffusion scheme according to Dudhia (1996). Finally, the surface layer physics are represented by the revised MM5 scheme after Jiménez et al. (2012). The combination of these physical parameterizations has been shown to adequately reproduce Mediterranean cyclones (e.g. Fita and Flaounas, 2018; Miglietta and Rotunno, 2019; Flaounas et al., 2021).

### 2.2 Numerical experiments

In order to study the characteristics of the upstream-downstream cyclone mechanism, we construct a semi-idealized atmospheric basic state, which corresponds to the climatological atmospheric conditions in the four seasons. We calculate it by taking the average of ERA5 reanalysis (Hersbach et al., 2020) at 0000 UTC and 1200 UTC during the period from 1979-2020. Initializing the model with climatological conditions, we remove synoptic-scale wave activity that provides the intense daily and sub-daily variability that characterizes the large-scale dynamics in the North Atlantic storm tracks. Further, it allows us to provide a more straightforward analysis of the relationship between North Atlantic cyclones and downstream cyclogenesis in the Mediterranean. Therefore, our semi-idealized numerical framework, with a climatological basic state and idealized initial perturbations, produces realistic large-scale dynamics in the region, without being influenced by already established wave activity. Note that this approach is not unlike the one used by Grams and Archambault (2016), who studied limited-area numerical simulations of extratropical transition in the western North Pacific, using initial and boundary conditions from compositing suitable large-scale flow situations. A major difference is that here, the initial conditions simply correspond to the seasonal mean state.





Figure 1 provides an overview of the climatological atmospheric state in the four seasons. Due to the averaging, the climatological PV fields at 300 hPa in Fig. 1a,d,g,j are lacking the sharp gradients we find in instantaneous fields. On the other

hand, averaging allows us to eliminate the intense daily and sub-daily variability of the atmospheric state and we can study the impact of imposed perturbations of the polar jet in the initial conditions. In DJF, the climatological fields show a prominent Icelandic Low (1000 hPa) and a broad Azores High (1020 hPa) that extends over the entire central North Atlantic into the Mediterranean and over North Africa (purple contours). In between those systems the polar jet spans from North America to the United Kingdom, with its maximum zonal velocity of about 40 m s$^{-1}$ at the east coast of North America (Fig. 1b).

To analyse the seasonal dependency of Mediterranean cyclogenesis with respect to large-scale atmospheric conditions, we also calculate the climatological atmospheric state for MAM, JJA and SON, see Fig. 1d-l. In both MAM and SON conditions the dynamical tropopause (depicted by the 2 PVU contour at 300 hPa; black line in Fig. 1b,e,h,k) is nearly at the same latitude, which in both cases is shifted towards the north compared to DJF, stronger for SON (Fig. 1k) than for MAM (Fig. 1e). The dipole of the Icelandic Low and the Azores High is weakest in MAM, moderate in SON and strongest in DJF. In MAM the

maximum zonal jet at 300 hPa (Fig. 1e) is at a similar position as in DJF, whereas in SON it is shifted northward and located around the Gulf of Saint Lawrence (Fig. 1k). In both MAM and SON, the jet is much weaker than in DJF, with maximum zonal wind speeds of slightly above 20 m s$^{-1}$. In MAM the jet becomes asymmetric to the south in the upper atmosphere (Fig. 1f) compared to SON (Fig. 1l) and DJF (Fig. 1c). In JJA both the dipole in SLP (Fig. 1g) and the polar jet, which primarily extends zonally at 50°N (Fig. 1h), are of the lowest amplitude (1010-1020 hPa and 15 m s$^{-1}$). The dynamical tropopause at 300 hPa

is entirely north to 60°N.

Model integration is initialised with the reference atmospheric states in Fig. 1, and localized wind, temperature and geopotential perturbations. These perturbations correspond to a positive upper-level quasi-geostrophic PV anomaly located at the latitude of the maximum zonal jet speed at 300 hPa. The anomalies are of different amplitudes ($A$), with a horizontal extension of $L_x = L_y = 1000$ km, and a vertical exponential decay length of $L_z = 4$ km, which have been used before in the studies by

Wernli et al. (1999) and Schemm et al. (2013):

$$\text{QGPV}(x,y,z) = A \left[ \exp\left( -\frac{x}{L_x} \right) \exp\left( -\frac{y}{L_y} \right) \exp\left( -\frac{z - z_\text{h}}{L_z} \right) \right], \tag{1}$$

where $z_\text{h}$ marks the height of the center of the anomaly, for which we use the geopotential height at the location of the maximum zonal jet. PV anomalies have been inverted according to the approach of Sprenger (2007), where we use a constant surface pressure $p_0 = 1017$ hPa, a uniform stratification in the troposphere (up to 10 km) of $N = 0.015$ s$^{-1}$, and constant potential

temperature at the surface of $\theta_\text{surface} = 290$ K, which provides the corresponding wind, temperature and pressure anomalies. From those anomalies we calculate the resulting geopotential height perturbation, $Z'$, at vertical level $k$ according to:

$$Z'_k = \frac{R}{g} \frac{T'_{k+1} + T'_k}{2} \log\left( \frac{p_k + p'_k}{p_{k+1} + p'_{k+1}} \right), \tag{2}$$

with the specific gas constant of dry air $R$ and the gravitational acceleration $g$. Here, the $'$ indicates the perturbation by the imposed anomaly. We assume that the surface ($k = 0$) is unaffected by the anomaly and add the perturbations to the reference

atmospheric state to obtain the initial conditions for our runs. We use quasi-geostrophic PV amplitudes of $A = 0.7$, 1.4 and

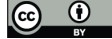


**Figure 1.** Overview of the climatological atmospheric conditions in DJF (a-c), MAM (d-f), JJA (g-i) and SON (j-l). (a,d,g,j): PV at 300 hPa, the purple contours show SLP with intervals of 5 hPa; (b,e,h,k): Zonal wind at 300 hPa, the black line shows the 2 PVU contour; (c,f,i,l): Vertical cross section of zonal wind along grey line in (b,e,h,k) from south to north. The 2 PVU contour is shown in purple.



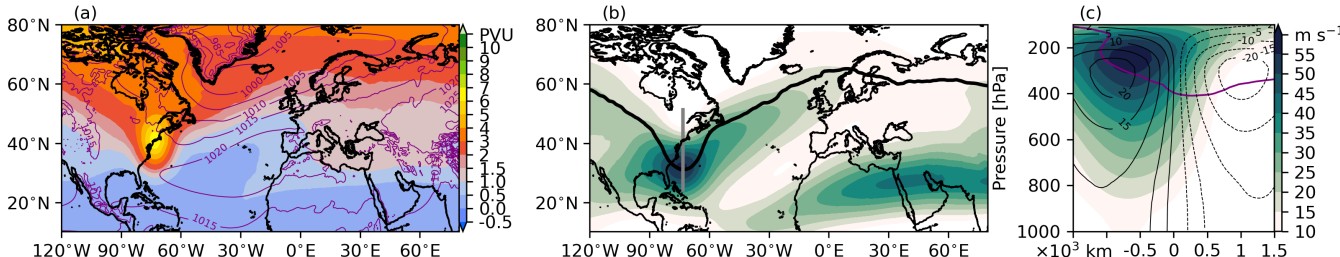

**Figure 2.** Same as Fig. 1a-c, but here with an additional upper-level positive PV perturbation with a $22\,\mathrm{m\,s^{-1}}$ wind anomaly. The black contours in (c) indicate the difference to the unperturbed reference state, solid lines show positive and dashed lines negative differences.

**Table 1.** Overview of simulations performed. The location indicates the horizontal position of the initial perturbation relative to the maximum zonal jet at 300 hPa, which is either central (C, no shift) or shifted by 200 or 400 km to the east, north, south and west. $^+$ indicates that 10 SPPT experiments were performed with the same initial conditions. $^*$ indicates simulations that are not explicitly discussed as they provided no additional insight.

| Season | location | maximum amplitude of the anomaly $[\mathrm{m\,s^{-1}}]$ | | | | | | | | |
|--------|----------|------|------|------|------|------|------|------|------|------|
| DJF | C | $5^{*,+}$ | $8^{*,+}$ | $11^+$ | $14^*$ | $17^*$ | $22^+$ | $25^*$ | $34^+$ | $45^*$ |
| | 200 km | | $8^*$ | 11 | | | $22^+$ | | $34^+$ | |
| | 400 km | | $8^*$ | 11 | | | $22^+$ | | $34^+$ | |
| MAM | C | | | $11^+$ | | | $22^+$ | | $34^+$ | $45^*$ |
| | 200 km | | | 11 | | | 22 | | 34 | |
| | 400 km | | | 11 | | | 22 | | 34 | |
| JJA | C | | | 11 | | | 22 | | 34 | $45^*$ |
| | 200 km | | | 11 | | | 22 | | 34 | |
| | 400 km | | | 11 | | | 22 | | 34 | |
| SON | C | | | $11^+$ | | | $22^+$ | | $34^+$ | $45^*$ |
| | 200 km | | | 11 | | | 22 | | 34 | |
| | 400 km | | | 11 | | | 22 | | 34 | |

$2.1 \times 10^{-4}\,\mathrm{s^{-1}}$, which translate to a maximum zonal wind speed perturbation of 11, 22 and $34\,\mathrm{m\,s^{-1}}$, respectively. Note, that hereafter, we will refer to the amplitude of the induced wind speed perturbation when mentioning the anomaly.

As an example, Fig. 2 shows the climatological atmospheric state in DJF that is perturbed by a quasi-geostrophic PV anomaly with a maximal amplitude in the zonal wind of $22\,\mathrm{m\,s^{-1}}$ at 300 hPa. The PV anomaly amounts to 4-5 PVU, resulting in a total
PV value of 6-7 PVU (Fig. 2a) at the location of the zonal jet maximum ($\approx 60\,\mathrm{m\,s^{-1}}$) at 300 hPa over the east coast of North America (Fig. 2b). Furthermore, the addition of the anomaly yields a deformation of the jet, i.e. an acceleration of winds in the south (solid contours in Fig. 2c).





For the experiments in Table 1 marked with a $^+$ we further perform a set of 10 ensemble simulations (using the same initial conditions) and utilize the stochastically perturbed parameterization tendency (SPPT) option in WRF to investigate the sensitivity of the mechanism to uncertainties in the parameterization of physical processes. These processes are important for the diabatic ridge amplification, e.g., in the outflow of WCBs and thereby potentially affect the RWB and downstream cyclogenesis (Gray, 2006; Spreitzer et al., 2019; Portmann et al., 2020). Note that the SPPT experiments in MAM and SON are performed for a reduced set of experiments according to Table 1, as they show the same variability of the cyclone dynamics as the ones performed for DJF (discussed in Sect. 5.2).

## 2.3 Cyclone tracking

To establish the relationship between cyclones, we apply the method of Wernli and Schwierz (2006) with the modifications described in Sprenger et al. (2017) to objectively identify and track cyclones in all simulations. We track the North Atlantic cyclone initialized by the imposed initial perturbation and the consequent Mediterranean cyclone. Note that each simulation will provide a pair of a North Atlantic cyclone (triggered by the initially imposed perturbation) and a Mediterranean cyclone (triggered by the PV streamer associated with the RWB that is induced by the North Atlantic cyclone).

# 3 The upstream-downstream cyclogenesis mechanism

## 3.1 Simulation with the unperturbed winter basic state

In this section we investigate how the large-scale circulation and cyclones develop in a simulation with unperturbed initial conditions. The left panels in Fig. 3 show the evolution of PV at 300 hPa and SLP in the first 8 days of the simulation. Two days after the model initialization (Fig. 3b), almost no upper-level wave activity developed and the large-scale circulation is still fairly close to the initial conditions (Fig. 3a). By day 4 (Fig. 3c), a (deep) cyclone starts to develop over the central North Atlantic while another weak one has reached Brittany (purple star), which formed at day 2 in the central North Atlantic and moved eastward (grey line in Fig. 3b-e). To the east of both cyclones, RWB is starting to develop, over western Europe and in the central North Atlantic, respectively, and by day 6 (Fig. 3d), both RWB events resulted in the intrusion of stratospheric PV streamers into the Mediterranean. The PV streamer over the central Mediterranean breaks anticyclonically, similarly to the so-called "life cycle 1" (LC1) in Thorncroft et al. (1993), and extends over northwest Africa. The second one is located over the Black Sea and Turkey. The streamers are separated by a narrow ridge over Greece, that is caused by the shallow cyclone, now located over Poland and Belarus (purple star in Fig. 3d). After 6.5 days, genesis of a very shallow Mediterranean cyclone (1015 hPa) took place close to Cyprus (grey line in Fig. 3e). As cyclogenesis took place underneath the PV streamer that resulted from the shallow North Atlantic cyclone, we associate this Mediterranean cyclone to the early shallow cyclone over the North Atlantic. After 8 days (Fig. 3e), the polar jet shows a pronounced wave activity with multiple ridges over the Mediterranean and North Atlantic, cyclones over the North Atlantic (south of Greenland, to the east of the UK, and of the





east coast of North America) that reach spatial sizes, amplitudes and intensities comparable to observed systems represented in reanalyses.

Compared to the cyclone climatology, the first cyclones that develop in this simulation over the North Atlantic and the Mediterranean, respectively, are of low intensity. Without a well-defined upper-level trough as initial perturbation, cyclogenesis is expected to be naturally weaker. Therefore, the North Atlantic cyclone starts with a low intensity of 1018 hPa and reaches its mature stage (time of minimum SLP along the cyclone track) over central Europe around day 5 with a central SLP of 1002 hPa. Also, the horizontal PV gradients along the PV streamer over the Black Sea and Turkey are weak and the resulting Mediterranean cyclone is shallow with 1015 hPa at cyclogenesis and 1010 hPa at its mature stage over Turkey. Later in the simulation, when upper-level waves have grown to larger amplitude, cyclones with more realistic intensity and sizes develop (Fig. 3d,e).

### 3.2 Simulation with an additional initial upper-level PV perturbation

In this section, we analyze how the presence of a North Atlantic cyclone, triggered by an initial upper-level PV perturbation (Fig. 2) at the location of the maximum climatological zonal jet at 300 hPa, affects the upper-level wave and downstream cyclogenesis. The positive PV perturbation has a wind amplitude of $22\,\mathrm{m\,s^{-1}}$ and represents an approaching trough over the North Atlantic (Fig. 3f). Compared to the evolution in the unperturbed simulation (Fig. 3a-e), the presence of the perturbation triggers almost instantaneous cyclogenesis over the North Atlantic, to the east of the perturbation. After 2 days of model integration (Fig. 3g), the cyclone has deepened below 985 hPa, while moving northeastward (grey line) to the south of Greenland (purple star). Furthermore, the perturbation led to the formation of an intense ridge and a disruption of the zonal orientation of the tropopause (Fig. 3g), which results in the formation and southward intrusion of a PV streamer to the west of the UK (Fig. 3g). The upper-level trough and ridge pattern, and the North Atlantic cyclone are spatially larger and more intense compared to the ones forming between days 4-6 in the unperturbed simulation (Fig. 3c,d). The North Atlantic cyclone continuously deepens until it reaches its mature stage between Greenland and Iceland, with a minimum SLP of 977 hPa around day 3. After day 4 (Fig. 3h), a prominent and, compared to the unperturbed reference run (Fig. 3c), narrow but large-amplitude PV streamer ($\approx$ 6 PVU) has intruded the western and central Mediterranean. The spatial size and amplitude of the streamer is comparable to PV streamers described in real case studies (e.g. Tripoli et al., 2005; Argence et al., 2008; Wiegand and Knippertz, 2014, their Fig. 4a, 1c and 8b, respectively), which induced intense Mediterranean cyclones. The streamer triggers cyclogenesis over Libya (grey line in Fig. 3h), from where the Mediterranean cyclone propagates to the northeast and transitions to the Mediterranean Sea northwest of Egypt on day 6 (Fig. 3i). At the time of minimum SLP around day 8 (Fig. 3j), the Mediterranean cyclone (994 hPa) is located over the eastern Pontic mountains. The location of cyclogenesis and lysis in our semi-idealized simulation are consistent with the extended winter and wet season climatology described by Almazroui et al. (2015) (their Fig. 3 and 4).

Perturbing the initial conditions at the center of the zonal jet immediately triggers cyclogenesis over the North Atlantic, with a prominent cyclonic RWB and a strong ridge forming downstream as a consequence. The timing, spatial size and intensity of the large-scale features, i.e., the cyclone over the North Atlantic, the ridge east of the cyclone, the intrusion of a PV streamer into the Mediterranean and Mediterranean cyclogenesis, are very different in the perturbed compared to the un-



**Figure 3.** Time evolution of the simulation initialized with the unperturbed climatological reference state in winter. Panels show PV at 300 hPa at (a) 0 d, (b) 2 d, (c) 4 d, (d) 6 d, and (e) 8 d of simulation time. Purple contours show SLP in 5 hPa intervals. The grey lines show the tracks of the initial North Atlantic and Mediterranean cyclones, and purple stars their current location.



perturbed experiment (Sec. 3.1). In particular, perturbing the zonal jet in this particular way reveals the upstream-downstream cyclogenesis mechanism. In the next section we discuss how different locations and amplitudes of the initial perturbation affect the wave evolution, the intensity and location of the North Atlantic cyclone, and the subsequent downstream development of the Mediterranean cyclone.

## 4  Sensitivity of the upstream-downstream mechanism to variations of the upstream conditions

We now probe the sensitivity of the upstream-downstream cyclogenesis mechanism in winter to different intensities and locations of the initial upper-level perturbation, and hence also of the North Atlantic cyclones. Therefore, we repeat the simulation in which the initial perturbation is positioned at the maximum intensity of the jet at 300 hPa (Fig. 3f-j), with two different amplitudes of the perturbation, i.e., 11 and 34 m s$^{-1}$, corresponding to half and 1.5 times the one applied in Sec. 3.2. Henceforth, we will refer to the three perturbation intensities as weak (11 m s$^{-1}$), moderate (22 m s$^{-1}$), and strong (34 m s$^{-1}$). Furthermore, we displace the perturbation by 200 and 400 km to the north, south, east and west, relative to the location of the maximum zonal wind at 300 hPa, which gives a total of 9 simulations per intensity of the perturbation (see Table 1). Regardless of the change in the location and intensity of the perturbation, the upstream-downstream mechanism is consistently occurring. In addition, by applying SPPT in selected simulations (see Table 1) we test whether the proposed mechanism is robust against small-scale atmospheric perturbations.

### 4.1  Morphology and intensity of the RWB

We first investigate the effect of displacing the initial perturbation on the RWB, for the moderate intensity case already discussed in Sec. 3.2. The effect on the PV streamer that forms on day 4 (Fig. 3h) is surprisingly small. Figure 4 shows the overlapping frequency (shading) of PV $\geq 2$ PVU at 300 hPa and the corresponding 2 PVU contours (colored lines) after 4 d for all 9 experiments in DJF with a moderate initial perturbation (without SPPT experiments). The dark red shading shows that the overall location and timing of the PV streamer over Central Europe are fairly consistent and rather independent of the imposed variations in the location of the initial upper-level PV perturbation. However, the exact position of the PV streamer is slightly shifted according to the horizontal shift of the initial perturbation, e.g. the eastward shifted perturbation results in an eastward shifted ridge and PV streamer (purple contour). The spatial displacement of ridges and PV streamers is slightly larger for a larger displacement of the initial upstream PV perturbation. Overall, similar results are found for the strong and weak initial perturbation, although the shifts in the position of the PV streamer over Europe are larger for strong perturbations compared to weak perturbations (not shown). This suggests that the variations of the initial upper-level PV structure over the North Atlantic are advected and further enhanced with the nonlinear flow evolution. This is consistent with the findings of Portmann et al. (2020), who showed that upstream differences in the PV structure over the Gulf of Saint Lawrence resulted in different shapes and location of PV streamers over the Mediterranean in an ensemble forecast. However, they observed shifts in longitude up to 15°, which is much larger than the shits observed in our experiments ($\pm 4°$). This difference might be due to the choice of the



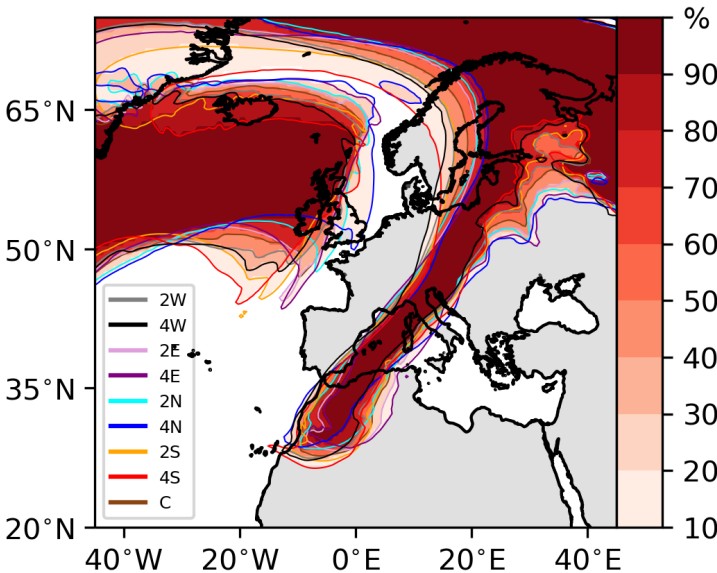

**Figure 4.** Averaged frequency of the PV streamer at $300\,\mathrm{hPa}$ ($\geq 2\,\mathrm{PVU}$) at $t = 4\,\mathrm{d}$ of the 9 simulations with the moderate intensity perturbation located at slightly different initial positions. Colored contours mark the $2\,\mathrm{PVU}$ line for the individual simulations. The legend indicates the horizontal shift relative to the location of the maximum zonal wind speed at $300\,\mathrm{hPa}$: labels 2 and 4 denote displacements by 200 and $400\,\mathrm{km}$, respectively, and E, N, W, and S indicate the direction of the shift. Finally, label C indicates the experiment with the perturbation located at the center of the jet (discussed in Sect. 3.2).

climatological initial conditions in our case, which appears to be more robust against small perturbations than instantaneous fields with larger isentropic PV gradients.

260    In contrast to the weak sensitivity to the location of the initial perturbation, the sensitivity to its initial amplitude (to the degree tested with our setup) is much larger. We first consider the evolution during the first $4\,\mathrm{d}$, i.e. until the formation of the PV streamer over Europe, in the weak and strong perturbation simulations (Fig. 5) and compare them with the simulation with the moderate perturbation (Fig. 3f-h). For all three simulations, the initial perturbation was not shifted relative to the position of the maximum zonal wind at $300\,\mathrm{hPa}$. Figure 5 shows that the stronger the initial perturbation, the larger becomes the ridge that

265    forms within $2\,\mathrm{d}$ over the central North Atlantic (Fig. 5a,c). Furthermore, in the experiment with a strong perturbation (Fig. 5c), the ridge has lower PV-values and its comma-shape extends further north. As a result, the cyclonic RWB occurs near southern Greenland compared to the central North Atlantic in the experiment with a weak perturbation (Fig. 5a). In the experiment with a strong perturbation, the more amplified RWB with a more intense negative PV anomaly in the ridge cause the intruding PV streamer over western Europe after $4\,\mathrm{d}$ to be of higher amplitude and to reach further south over Algeria (Fig. 5d), compared

270    to the weak perturbation experiment (Fig. 5b).

These important differences in the character of the RWB and structure of the PV streamer over Europe after $4\,\mathrm{d}$ when increasing the amplitude of the initial perturbation are still very obvious when comparing small ensembles for each perturbation



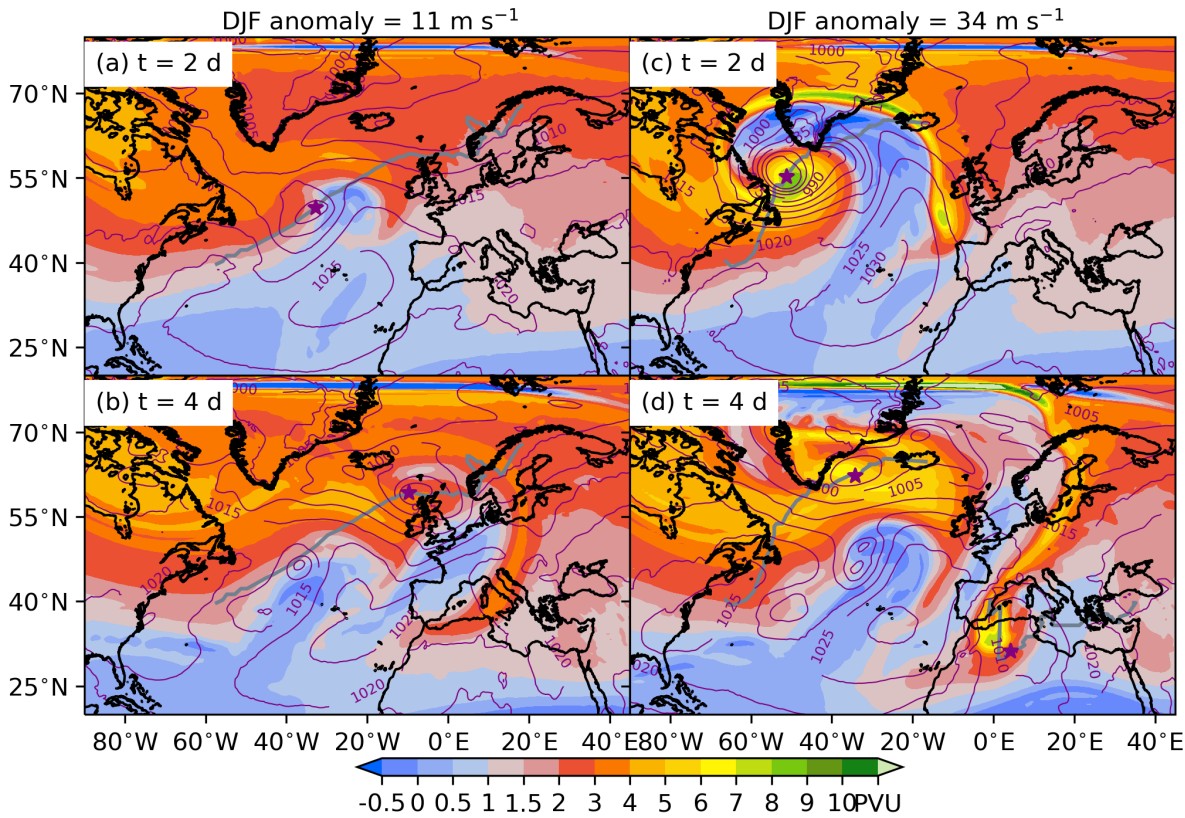

**Figure 5.** Same as Fig. 3 for an initial wind anomaly of $11\,\mathrm{m\,s^{-1}}$ (a,b) and $34\,\mathrm{m\,s^{-1}}$ (c,d). Panels show the development after 2 and 4 d, respectively.

amplitude. To this end, ensembles with 9 simulations each were performed for the weak, moderate and strong initial perturbation with slightly shifted positions of the initial perturbation as described above for the moderate intensity case. Figure 6 shows the ensemble averaged PV at 300 hPa for the three intensity categories, revealing well-defined PV streamers and therefore a very similar large-scale evolution for the simulations with the same intensity of the initial perturbation. With a weak initial perturbation (Fig. 6a), the PV streamer curves anticyclonically and extends over Italy. In the experiments with a moderate (Fig. 6b) and strong (Fig. 6c) initial perturbation, the PV streamer has higher PV values and extends further equatorward. In the moderate case, a straight PV streamer forms with almost uniform PV from Poland to Morocco, whereas in case of the strong perturbation, a "bulb" (or tip vortex) with high PV values forms at its southern end. The composite SLP fields (purple contours in Fig. 6) reveal that with increasing intensity of the initial upstream perturbation, the higher PV values of the PV streamers provide a stronger baroclinic forcing and thus cyclogenesis occurs earlier in the simulation (further discussed below). In the simulations with the strong initial perturbation (6c), a weak surface cyclone appears in the composite at day 4 over northwestern Africa, most likely induced by the prominent upper-level tip vortex.





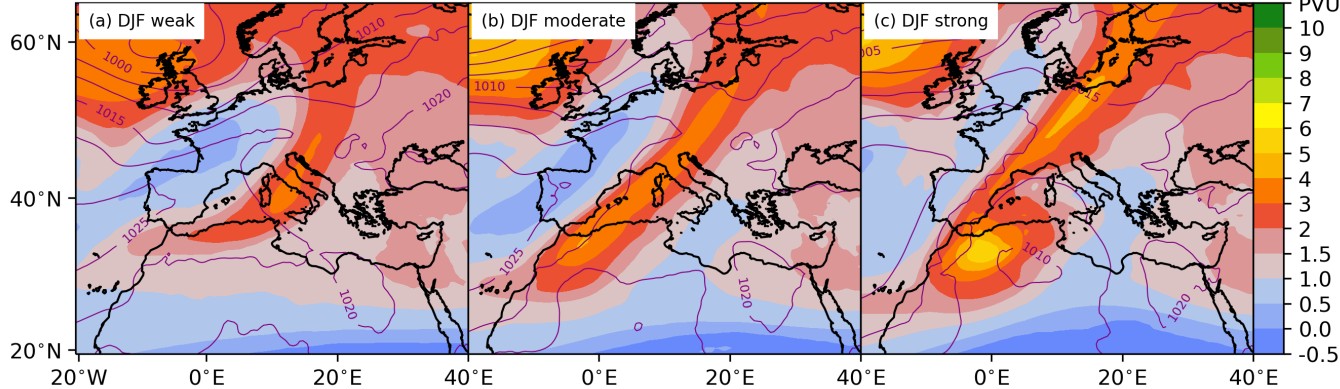

**Figure 6.** Composites after 4 d of 9 simulations each with slightly displaced initial perturbations (Table 1) for (a) "weak", $11\,\mathrm{m\,s^{-1}}$, (b) "moderate", $22\,\mathrm{m\,s^{-1}}$, and (c) "strong", $34\,\mathrm{m\,s^{-1}}$ perturbation amplitudes. Colors show PV at 300 hPa, and the purple contours SLP with 5 hPa intervals.

## 4.2 Tracks and intensities of the upstream and downstream cyclones

We now turn our attention to the effects of the initial perturbation on the resulting North Atlantic and Mediterranean cyclones, first considering cyclones in the 9 simulations in the moderate intensity category (Fig. 7). The North Atlantic cyclones (Fig. 7a) propagate fairly coherently ($\pm 2°$) from south of Newfoundland towards Iceland and reach the mature stage at similar locations (indicated by asterisks), except for the simulation labeled 4S, where the initial perturbation is shifted 400 km to the south. Larger differences occur at the end of the cyclone tracks. The Mediterranean cyclone tracks (Fig. 7c) show slightly more spread with substantial differences in the location of the mature stage, which varies from the Aegean Sea to Syria. Considering also the evolution of minimum SLP along the cyclone tracks (Fig. 7b) reveals substantial differences in the timing of the intensification and the maximum intensity reached for both the North Atlantic and Mediterranean cyclones. For instance, the peak intensity of the North Atlantic cyclone in the simulations 4N and 4S differs by 10 hPa and the time of maximum intensity by 2 d, and for the Mediterranean cyclones, there is a relatively poor agreement in the timing of the intensification.

As a next step, we consider the maximum intensity of North Atlantic and Mediterranean cyclones in all simulations listed in Table 1. The goal is to identify potential correlations between the intensities of the upstream-downstream cyclone pairs, and between characteristics of the initial conditions and the intensity of the North Atlantic cyclone. To this end, the maximum zonal wind at 300 hPa in the vicinity of the initial perturbation is chosen as an important characteristic of the initial conditions (via the thermal wind relationship, high wind speed implies strong baroclinicity). This zonal wind maximum clearly increases with the amplitude of the PV perturbation, but it also depends on the position of the perturbation. For instance, for the moderate intensity perturbation located at the central position, the maximum zonal wind in the initial conditions amounts to about $58\,\mathrm{m\,s^{-1}}$ (see Fig. 2b,c). Figure 8a shows that the maximum intensity (minimum SLP along the cyclone track) of the North Atlantic cyclone (grey dots) increases almost linearly with the maximum zonal jet velocity in the initial conditions (the correlation is



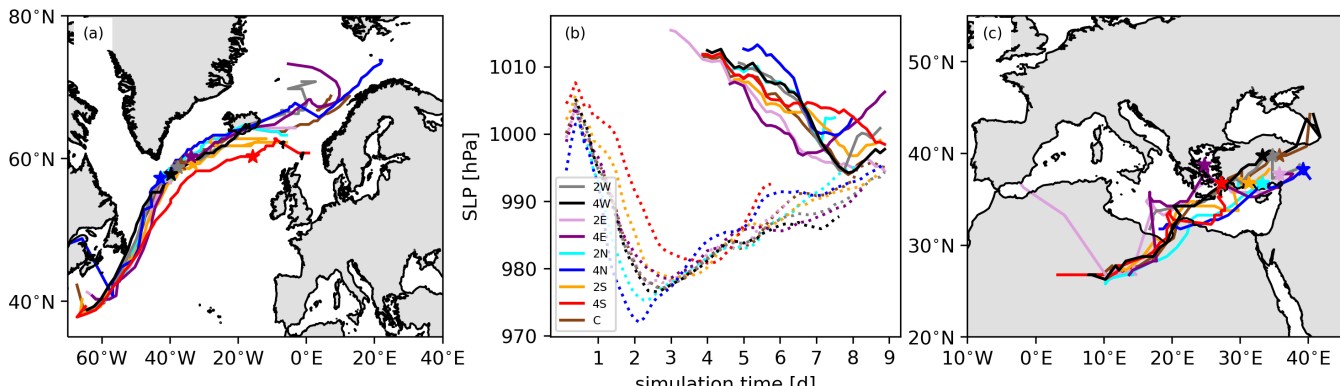

**Figure 7.** Cyclone tracks and characteristics for all non-SPPT experiments in winter with a perturbation strength of $22\,\mathrm{m\,s^{-1}}$. (a) North Atlantic cyclone tracks, maximum intensity highlighted with asterisks; (b) SLP evolution of the North Atlantic (dotted) and Mediterranean (solid) cyclones. The legend indicates the horizontal location of the initial perturbation relative to the location of the maximum zonal jet: 2 and 4 mark displacements by 200 and 400 km, respectively and E, N, W, S indicate the direction of the shift. C labels the experiment with the perturbation located at the center of the jet; (c) Mediterranean cyclone tracks, asterisks again highlight maximum intensity.

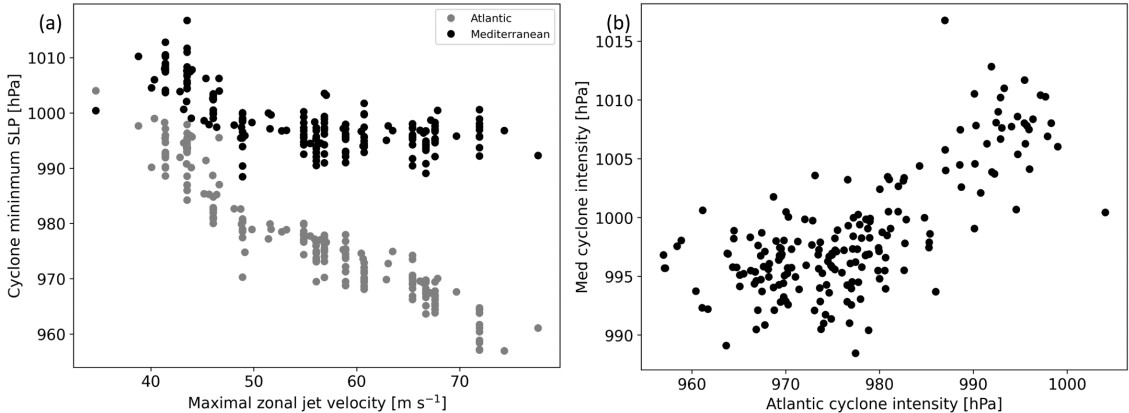

**Figure 8.** Scatter plots for all winter simulations of (a) the maximum 300 hPa zonal wind near the initial perturbation and the minimum SLP of the North Atlantic (gray) and Mediterranean (black) cyclones, and (b) the minimum SLP of the Mediterranean vs. the North Atlantic cyclone. Every dot represents one simulation, and this analysis also includes the SPPT experiments (see Table 1), which align in (a) at the same zonal wind value.

305 statistically significant with $\sigma = -0.93$). Perturbations with higher PV values induce a stronger cyclonic wind field for the vertical baroclinic coupling, similar to the results of Barnes et al. (2022). Furthermore, they create a more intense jet streak, leading to increased large-scale forcing for upward motion that amplifies the development of the cyclone.



The maximum intensity of the Mediterranean cyclone (black markers in Fig. 8a) also increases with an increased maximum zonal wind speed at 300 hPa, but the correlation is much less obvious than for the North Atlantic cyclone. There appears to be
a threshold zonal wind speed value of about $50\,\mathrm{m\,s^{-1}}$ beyond which the intensity of the Mediterranean cyclones remains fairly constant. The discussion above has shown that the PV streamers in the experiments with stronger perturbations have higher maximum PV values, which could lead to stronger Mediterranean cyclones. However, since other non-linear atmospheric processes also contribute to the intensity of the Mediterranean cyclone, there is no straightforward correlation with the amplitude of the initial maximum zonal upper-level wind as we found for the North Atlantic cyclones. The variability of the intensities
of the Mediterranean cyclones suggests that this intensity depends on the detailed structure and location of the PV streamer, which is consistent with the findings from several previous studies (Flocas, 2000; Romero, 2001; Homar and Stensrud, 2004; Argence et al., 2008; Claud et al., 2010; Carrió et al., 2017).

In Fig. 8b, we correlate the intensities of the cyclone pairs, which consistent with Fig. 8a shows a good correlation for relatively weak North Atlantic cyclones and no correlation for North Atlantic cyclones deeper than 980 hPa. The primary
finding of this analysis is that while the intensity of the North Atlantic cyclone is closely connected to the amplitude of the initial upper-level zonal wind peak, the connection is substantially weaker for the intensity of the Mediterranean cyclone downstream. Consistent with previous studies, the dynamics of the latter is heavily influenced by local factors like sea surface temperature and orography in the Mediterranean, and by the non-linear flow evolution during the 4-day simulation period prior to Mediterranean cyclogenesis.

## 5 Seasonal variability of the upstream-downstream mechanism

So far, all experiments were performed for (perturbed) winter-mean initial conditions. In this section, we investigate whether the upstream-downstream mechanism exists when initializing the simulations with climatological mean states of the other seasons. Thereby, we investigate how the track and intensity of the North Atlantic cyclone, the morphology of the RWB and the PV streamer, and the track and intensity of the Mediterranean cyclone are affected by different locations and intensities of the polar jet (Fig. 1), as well as different temperature and moisture profiles. Therefore, we repeat the previous experiments, but now use the climatological mean states for MAM, JJA and SON (see Table 1 and Fig. 1). Independent of the locations and amplitudes of the initial perturbation, the downstream cyclogenesis over the Mediterranean occurs in all simulations for MAM and SON, but not for JJA, which is consistent with the lowest observed cyclone frequency during the summer season (Campins et al., 2011; Flaounas et al., 2013; Lionello et al., 2016).

### 5.1 Morphology of the RWB

First, we investigate the evolution of the SON and MAM simulations that are perturbed by a moderate initial perturbation, placed at the maximum zonal wind speed of the polar jet at 300 hPa, similar to the DJF experiment in Sect. 3.2. Figure 9 shows the evolution of PV at 300 hPa and SLP for the SON (Fig. 9a-e) and MAM (Fig. 9f-j) initial conditions (compare to DJF in Fig. 3f-j). In SON, the initial perturbation is located over the Gulf of Saint Lawrence (Fig. 9a), whereas in MAM it is located over





**Figure 9.** Same as Fig. 5f-j for initial conditions corresponding to the climatological mean in SON (a-e) and MAM (f-j).





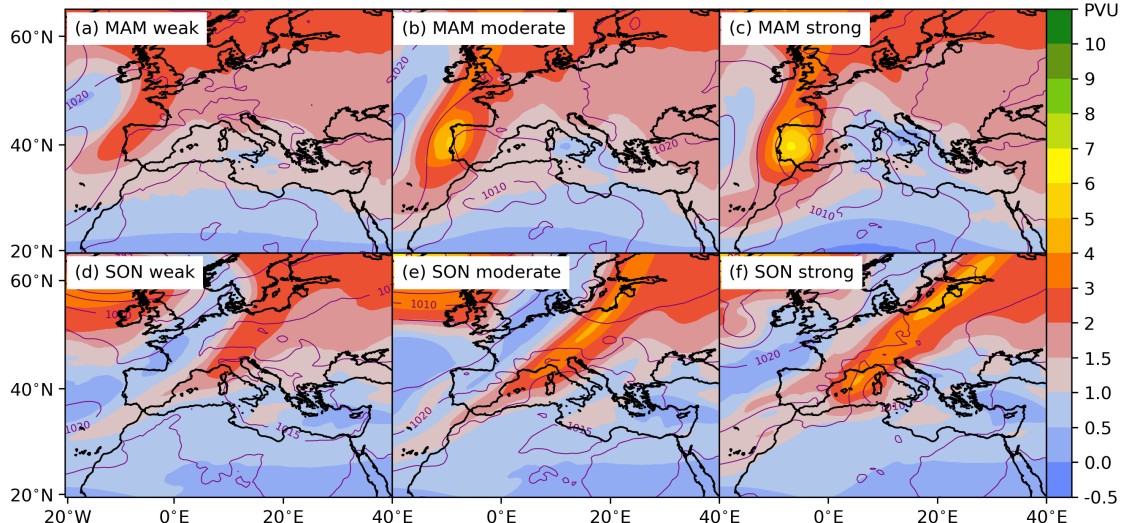

**Figure 10.** Same as Fig. 6 for initial conditions corresponding to the climatological mean in MAM (a-c) and SON (d-f).

the east coast of North America (Fig. 9f), similar to the one in DJF (Fig. 3f). With the more northerly location of the jet, and thereby the initial perturbation, at about 50°N in SON (compared to about 40 °N in DJF), cyclogenesis over the North Atlantic occurs at higher latitudes and the North Atlantic cyclone propagates primarily eastward (grey line in Fig. 9b) in the first 2 d, compared to the northeastward propagation of the cyclones forming close to Newfoundland in MAM (Fig. 9g) and DJF (Fig. 3g). After 4 days, both the ridge and PV streamer in SON are comparable in morphology to the ones in DJF (Fig. 3h), but the

PV streamer over the Mediterranean has lower PV values (Fig. 9c). Over the North Atlantic, the ridge, the RWB and the PV streamer are all located further north, with the tip of the PV streamer located over the Gulf of Genoa (Fig. 9c), in contrast to the DJF simulation where the tip reaches Algeria (Fig. 3h). In SON, the PV streamer triggers Mediterranean cyclogenesis in the Tyrrhenian Sea, from where the cyclone moves eastwards over Greece (purple asterisk in Fig. 9d) and afterwards into the Black Sea (Fig. 9e).

Repeating the experiment with the climatological mean state in MAM, reproduces the upstream-downstream mechanism with a similar development of the ridge, RWB and PV streamer (Fig. 9g,h) as in DJF (Fig. 3g,h). However, the intrusion of the PV streamer differs as it is located further west, over Portugal and Spain after 4 days (Fig. 9h). It appears that the weaker jet in MAM (Fig. 1e) leads to a slower eastward propagation of the flow compared to the more intense jet and thereby faster propagation in DJF. The intrusion of the PV streamer into the western Mediterranean triggers cyclogenesis over Algeria, from

where the cyclone propagates northwards into the Balearic Sea (purple asterisk in Fig. 9i) and afterwards into the Adriatic Sea (Fig. 9j).

These results showed that for all MAM and SON experiments with a moderate initial perturbation we find similar RWB patterns and coherent locations of the PV streamers after 4 d of model integration, similar to DJF (Fig. 4). The average locations of the PV streamers of the 9 ensemble experiments are summarized in Fig. 10 for weak, moderate and strong initial





perturbations. For all perturbation amplitudes, the MAM PV streamer is located over Spain and Portugal (Fig. 10a-c), while in SON it is located over the Gulf of Genoa (Fig. 10d-f) and the Balearic Sea, compared to Northern Africa and Algeria in DJF (Fig. 6). For MAM, Fig. 10 shows that the larger the amplitude of the initial perturbation the more the tip of the PV streamer tends to wrap cyclonically, resulting in strongly positive PV anomalies in the upper troposphere, similar to DJF (Fig. 6). In all simulations the stronger baroclinic forcing by higher-amplitude PV streamers provokes earlier cyclogenesis, as indicated by
the composite SLP fields (purple contours in Fig. 10c,f).

## 5.2   Cyclone tracks and intensities of the upstream and downstream cyclones

In this section, we focus on the change in the tracks and intensities of the North Atlantic and Mediterranean cyclones, similar to Sect. 4.2. Note that since the variability of the cyclone tracks and their intensities in the experiments in MAM and SON in the 9 experiment ensemble is comparable to the findings presented in Sect. 4.2, we now include all experiments (also those
with SPPT; see Table 1) in the final analysis.

    Figure 11 shows the cyclone track frequency (percentage of simulations that share cyclone track points within $\pm 2.5°$) of the North Atlantic and Mediterranean cyclones. The similar morphology and location of the polar jet in DJF and MAM leads to comparable cyclone tracks over the North Atlantic, with cyclogenesis taking place south of Newfoundland both in DJF (11a-c) and MAM (11d-f). In SON, the North Atlantic cyclones form over the central North Atlantic (Fig. 11g) and east of
Newfoundland (11h,i), as the polar jet is located at higher latitudes (Fig. 1k). Independent of the season, the North Atlantic cyclones propagate more zonally in the experiments with weak (Fig. 11a, d, g) and and more meridionally in the simulations with strong initial perturbations (Fig. 11c,f,i). The high track frequencies (dark red) in each panel of Fig. 11 suggest that the cyclone tracks over the North Atlantic are fairly coherent, independent of the initial upstream upper-level PV perturbation and of random small-scale perturbations induced by SPPT. The deepest cyclones over the North Atlantic form in DJF and SON,
where the baroclinicity is larger compared to MAM. The deepest North Atlantic cyclones in spring reach intensities of about 990 hPa compared to 970 hPa in DJF and SON (not shown).

    The different location of the PV streamers (Fig. 10 and Fig. 6) leads to Mediterranean cyclogenesis in different regions. In DJF (Fig. 11a,b,c), cyclogenesis takes place over Libya (Fig. 11a,b) from where the cyclones propagate towards Greece, Cyprus and Turkey. In MAM cyclogenesis takes places over Algeria for the weak and moderate initial perturbations, from
where the cyclones move over to the east and west of Italy, respectively (Fig. 11d,e), where cyclogenesis occurs in case of the strong perturbation experiments. In SON cyclogenesis takes place in the central Mediterranean and the cyclones propagate eastwards (Fig. 11g,h,i). In all experiments with a strong initial perturbation (Fig. 11c,f,i), the PV streamers have larger PV values (Fig. 6 and Fig. 10c,f) and thus provide a stronger baroclinic forcing that triggers Mediterranean cyclogenesis at earlier times (Fig. 12c,f,i), resulting in westward shifted cyclogenesis and cyclone tracks (Fig. 11c,f,i). The Mediterranean cyclone
tracks in DJF are consistent in time and space with the tracks of Almazroui et al. (2015) during the extended winter period. The tracks and mature stages of Mediterranean cyclones in MAM and SON are consistent with the tracks from Campins et al. (2006). However, cyclogenesis does not take place in the hotspots found by Bartholy et al. (2009) in either season. Note that in DJF, there is a larger variability in the cyclone tracks due to the larger number of SPPT experiments (Table 1). This spread





**Figure 11.** Cyclone track density for all DJF (a-c), MAM (d-f), SON (g-i) experiments, initialized with (a,d,g) weak, (b,e,h) moderate, and (c,f,i) strong initial perturbation experiments.



**Figure 12.** SLP evolution of Mediterranean cyclones in all (a-c) DJF, (d-f) MAM, and (g-i) SON simulations, perturbed with a (a,d,g) weak upper-level PV anomaly, (b,e,h) moderate-intensity anomaly experiments, and (c,f,i) strong PV anomaly. Boxes mark the interquartile range, whiskers the 10th and 90th percentile and the red and black line the mean and median, respectively.

becomes most visible for experiments with strong initial perturbations (Fig. 11c,f,i). Here, the amplitude of diabatic processes
is larger due to the larger amplitude of the PV streamer and therefore the variability provoked by the SPPT is increased.

The different seasonal-mean conditions over the Mediterranean and the different intrusion of the PV streamers not only result in different locations of cyclogenesis but further provide different intensification rates and maximum intensities of Mediterranean cyclones. Figure 12 shows the distribution of SLP along the Mediterranean cyclone tracks. In all experiments, the average SLP (red) shows a steeper decrease and reaches lower minimum values with increasing amplitude of the initial per-
turbation (columns in Fig. 12), suggesting that the Mediterranean cyclones obtain a stronger maximum intensification and




maximum intensity. This is consistent with the higher-amplitude PV streamers, which provide a stronger baroclinic forcing and thus a stronger intensification of the Mediterranean cyclones. The increase in the intensity of Mediterranean cyclones is similar for all experiments in all seasons, except for MAM in the experiments with a strong initial perturbation (Fig. 12f), where the Mediterranean cyclones obtain an average minimum of about 996 hPa that is about 4 hPa lower than for the ones

in the experiments with an moderate initial perturbation (about 992 hPa in Fig. 12e). Although the cyclones are weaker in the experiments with a strong initial perturbation in MAM, they maintain their maximum intensity for about 3 d, as suggested by the increasing perseverance of lowest average SLP value (red in Fig. 12f). This might be due to less propagation of the cyclones over the Balearic Sea (Fig. 11f) where they access the moisture provided by ocean surface and therefore can maintain their intensity by latent heat release in convection. Note, that the spread of SLP in MAM and SON is generally lower than in DJF,

due to the fewer SPPT simulations performed for those seasons, which provide the largest variance.

Compared to DJF and SON, we find the most intense Mediterranean cyclones in MAM, which is in agreement with the seasonal intensity of Mediterranean cyclones. Here, higher sea surface temperatures (SSTs) and a less stable atmospheric state, i.e., weaker vertical temperature gradients, might favor convection and thus result in a stronger deepening of the Mediterranean cyclones. Why high-amplitude PV streamers in experiments with strong initial perturbations do not further increase the in-

tensity of the Mediterranean cyclones is not straightforward to explain. Considering the results of Flaounas et al. (2021), who showed that the stronger the upper-level forcing to the intensity of Mediterranean cyclones the weaker the diabatic one, there might be a balancing factor that makes it difficult to attribute a straightforward connection between the amplitude of the PV streamers and the intensity of the cyclones. Here, very strong convection might erode the upper-level PV and thereby inhibit its maximum contribution to the intensity of the cyclones. However this would need another detailed analysis and comparison

of those experiments.

## 6   Discussion and conclusion

For the first time, we show, using a semi-idealized simulation setup, the direct relationship between upstream North Atlantic and downstream Mediterranean cyclones, which we refer to as the upstream-downstream mechanism and investigate its sensitivity to variations in the upstream PV perturbation. Therefore, we introduce a modeling framework using WRF in which

we prescribed the climatological seasonal-mean atmospheric states as initial and boundary conditions and perturbed them by a weak, moderate and strong PV anomaly near the maximum zonal wind speed of the polar jet at 300 hPa. We investigated the flow evolution over the North Atlantic and Mediterranean region, which is primarily affected by the North Atlantic cyclone that forms rapidly after the model initialization due to the presence of the initial perturbation. An upper-tropospheric ridge builds up downstream of the cycle and thereby provokes RWB, causing a PV streamer to intrude the Mediterranean and trigger

cyclogenesis or enhance the intensity of a Mediterranean cyclone after about 4 d of simulation time.

To investigate the sensitivity of this upstream-downstream mechanism to variations in the upstream upper-level PV structure over the North Atlantic, we changed both the horizontal position of the initial perturbation around the maximum zonal jet and its amplitude. Our results show that the upstream-downstream cyclogenesis mechanism occurs consistently, regardless of the the



position and amplitude of the initial perturbation (within the variations considered in this study). Furthermore, we find that the

intensity of the synoptic-scale dynamics, i.e., the intensity of the North Atlantic cyclone, the area and amplitude of the ridge, the

amplitude of the PV streamer over the Mediterranean and thereby, to some extent, the intensity of the Mediterranean cyclone,

are strongly influenced by and correlate with the amplitude of the initial upper-level PV perturbation. With increasing amplitude

of the initial perturbation we find a northward shift of the North Atlantic cyclone track and the RWB, but a similar position of

cyclogenesis in the Mediterranean in a given season. This suggests, that different flavors of RWB and resulting structures of

PV streamers can lead to cyclogenesis in similar regions in the Mediterranean. Furthermore, results from simulations including

SPPT show that the proposed mechanism is robust against random perturbations of model physics, which have the potential to

affect the formation of the ridge and thereby the RWB (Gray, 2006; Spreitzer et al., 2019).

To study the influence of the differing seasonal large-scale flow configurations on the upstream-downstream mechanism,

we repeated the experiments using as initial conditions the climatological seasonal-mean conditions of MAM, JJA, and SON.

The upstream-downstream mechanism is not present in the simulations for JJA, which is plausibly consistent with the lowest

cyclone frequency during that time period (Flaounas et al., 2013, 2015). For MAM and SON, the RWB shows a similar

spread as in DJF and our findings demonstrate that the seasonal position of the polar jet influences the specific location of the

intrusion of PV streamers to the Mediterranean. Thereby, cyclogenesis is provoked in different regions of the Mediterranean

(e.g., preferentially in the eastern Mediterranean when using DJF initial conditions and further west when using SON and

MAM initial conditions), with the corresponding seasonal variability in the tracks and intensity of these cyclones.

The tracks of the North Atlantic cyclone as well as the RWB are fairly insensitive to horizontal shifts of the initial perturba-

tion by 200 and 400 km. However, this lack of sensitivity of the North Atlantic cyclone tracks and downstream intrusion of the

PV streamer to the initial perturbation might be due to our choice of the climatological atmospheric states as initial conditions.

For instance, Portmann et al. (2020) showed that the detailed structure of a high-PV anomaly in the lower stratosphere over

the Gulf of Saint Lawrence (north to the jet streak) in an ensemble forecast resulted in different shapes and location of PV

streamers over the Mediterranean prior to cyclogenesis and consequently different cyclone tracks and intensities. Therefore,

the coherency of the downstream PV streamer in our simulations might be the result of weaker gradients at the level of the

tropopause (because we consider climatologically smoothed initial conditions), which might be less sensitive to perturbations

than sharper gradients in instantaneous fields. However, this choice of initial conditions allows us to establish a clear connection

between the simulated North Atlantic and Mediterranean cyclones.

The small shifts of the location of the PV streamers over the Mediterranean, due to variations in the horizontal positions of

the initial perturbation, and the associated changes in their upper-level PV structure affect the intensity of the Mediterranean

cyclones. The weaker the amplitude of the PV streamer the more essential its PV structure becomes for the intensity of the

Mediterranean cyclone. This sensitivity of the intensity of Mediterranean cyclones to the upper-level PV structure is consistent

with case studies of heavy-precipitation events and Mediterranean cyclones (e.g. Fehlmann and Davies, 1997; Fehlmann and

Quadri, 2000; Romero, 2001; Homar and Stensrud, 2004; Argence et al., 2008; Chaboureau et al., 2012). They showed that

even mesoscale variations in the upper-tropospheric PV structure resulted in different intensities (and tracks) of those events. In



essence, our results show that the weaker the initial upstream perturbation, the more the intensification and maximum intensity of the North Atlantic and Mediterranean cyclone are affected by changes in the upstream upper-level PV structure.

Nevertheless, factors such as seasonal variations of static stability and Mediterranean SST, and the relative location of cyclones to orography might further contribute to the variability of the intensity and tracks of the Mediterranean cyclones. In particular in the experiments during MAM the intensity of the downstream Mediterranean cyclone is rather independent of the intensity of the initial upstream perturbation, in contrast to the experiments in DJF and SON, in which the intensity of the Mediterranean cyclone scales to some extent with the amplitude of the initial upstream perturbation. This suggests that the

atmosphere in MAM might have reduced static stability and thus favors convection and intensification of the Mediterranean cyclone provoked by the approaching PV streamer, but also that the intensity of the Mediterranean cyclones is significantly affected by the temperature and moisture distribution over the Mediterranean. In this regard, although we found that increasing the horizontal resolution to $0.1°$ does not significantly change the variability of the intensities of the Mediterranean cyclones nor their tracks, future studies could address the role of convection to the intensity of the cyclones in experiments with higher

resolution and explicitly resolved convection.

     In our study, we investigated the impact of North Atlantic cyclones on downstream cyclone dynamics over the Mediterranean. We specifically analyzed how the intensity and location of North Atlantic cyclones affect this mechanism. However, our research is restricted to the selected initial conditions that impact this link. Therefore, identifying pairs of North Atlantic and Mediterranean cyclones in ERA5 reanalyses, similar to Raveh-Rubin and Flaounas (2017), could serve to further study the

relationship between the position and strength of North Atlantic cyclones and subsequent Mediterranean cyclones. Such an analysis could contribute to research about the correlation between cyclone properties in the North Atlantic and the resulting cyclogenesis in different regions of the Mediterranean, with interesting implications for forecasting Mediterranean cyclones. Additionally, the proposed modeling framework provides a new perspective on the future trends of Mediterranean cyclones. Recent studies (e.g Lionello and Giorgi, 2007; Ulbrich et al., 2009; Cavicchia et al., 2014; and Nissen et al., 2014) have

examined the impact of future climate projections on the tracks and intensities of Mediterranean cyclones. However, the interpretation of the spread of various model results requires support from a more comprehensive understanding of large-scale forcing of Mediterranean cyclogenesis, specifically the effects of future changes in the upstream-downstream mechanism. Presently, this aspect is missing from the state of the art and is a critical factor for comprehending the impact of climate change on regional high-impact weather.

*Data availability.* The WRF model and ERA5 reanalysis are openly available. Upon request the authors provide the namelists used to perform the simulations.

*Author contributions.* AS prepared all analyses and the manuscript. EF and HW provided scientific advice throughout the project and provided valuable suggestions for improving the manuscript.



*Competing interests.* Heini Wernli is executive editor of WCD.

*Acknowledgements.* AS acknowledges funding form the Swiss National Science Foundation (Project 188660). Furthermore, AS thanks Lukas Papritz for the help with the simulation setup and Michael Sprenger for guidance with the cyclone tracking. This work is a contribution to the COST Action CA19109 "MedCyclones: European Network for Mediterranean Cyclones in weather and climate".





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
