# Peer review of "The upstream-downstream connection of North Atlantic and Mediterranean cyclones in semi-idealized simulations"

_EGUsphere, 2023_

## Referee Comment (RC1)

**Review of 'The upstream-downstream mechanism of North Atlantic and Mediterranean cyclones in semi-idealized simulations'**

I found this manuscript to be very well written and easy to follow. The experiments were carefully designed and the analysis was thorough and insightful. I have a few substantive comments on the analysis and discussion and several recommendations for clarification, all of which should be fairly straightforward to address. Therefore, my recommendation is for the authors to make minor revisions prior to publication.

**General Comments:**

1. The only additional analysis I would to see is a comparison of pairs of cases (selected based on Fig. 8) for which the Med cyclone intensity is similar, but the Atlantic cyclone intensity is different. Specifically, I would like to know how the max jet speed compares between the two cases, and where in the process chain linking Atlantic cyclones to Med cyclones there is a breakdown that leads to the limited sensitivity of downstream cyclone intensity to the upstream cyclone. For example, do the PV streamers or RWB events differ substantially between the two cases, and if so, does this imply that these nonlinear links in the upstream-downstream process chain are key sources of downstream variability/uncertainty? Or are the differences between the two cases strictly local in the sense that difference growth primarily occurs within the Mediterranean region? There is a bit too much speculation around Line 315 on this matter, and I feel that this comparative analysis could help make this part of the manuscript less speculative.

2. The abstract states that the sequence of events leading to Mediterranean cyclogenesis "rarely occur in a spatially consistent, fully repetitive pattern", but the results from the idealized simulations actually suggest that this sequence of events leads to Mediterranean cyclogenesis rather consistently. Details of the intensity and position of the Mediterranean cyclone may vary by time of year or with the intensity of the upstream perturbation, but the overall sequence of events leading the cyclogenesis in the Mediterranean seems quite robust in the simulations. It may help to clarify in the abstract this apparent lack of consistency in the proposed mechanism is for real cyclone cases. Additional statements in the discussion about possible reasons why this pattern more reliably leads to cyclogenesis downstream in the simulations than in the real atmosphere would also be useful. For example, could the consistency of the mechanism in simulations be a result of using smoothed, climatological initial conditions?

3. I am curious if the fact that this mechanism is absent during JJA implies that it is contingent upon there being a zonally extensive and continuous jet stream waveguide across the Atlantic, or if the lack of downstream Mediterranean cyclogenesis during JJA is simply a result of weaker upstream baroclinicity over the North Atlantic preventing the formation of North Atlantic cyclone seeds. If the existence of the waveguide is a necessary condition for the upstream-downstream mechanism, then I would also imagine that climate change impacts on the jet stream/waveguide will influence this mechanism. For example, a weaker jet in future climates could make everything more JJA-like, in which case Med cyclones triggered through this mechanism might become less common. I understand that this is well beyond

the scope of the study, but it may be worth mentioning at the very end of the manuscript that climate change impacts on the jet stream could conceivably influence the prevalence of this mechanism in future climates.

**Clarification Comments**

Figure 3 caption: The caption should state what the difference between the left and right columns is.

Figure 3: I found the purple stars and cyclone tracks to be very difficult to see on these plots. Perhaps they could be outlined in white to enhance their visibility.

Line 186: It's somewhat ambiguous what "the second one" refers to here; consider replacing "one" with "PV streamer" for added clarity

Line 195: To what cyclone climatology are the authors referring here?

Figure 6 caption: Here the terminology is "perturbation amplitude", but in the Fig. 5 caption, it is referred to as a "wind anomaly" and in earlier captions it is "perturbation intensity." I recommend being consistent in the text and captions regarding the terminology for the strength of the perturbation to the jet.

Line 305: Please state in the text the significance test and confidence level for assessing the statistical significance of this correlation.

Line 322: Here the authors state that the connection between Med cyclone intensity and the max upper-level jet velocity is "substantially weaker" than the connection between the Atlantic cyclone intensity and the max upper-level jet velocity. This seems an overly-simplistic statement that overlooks the fact (acknowledged in the previous paragraph) that the connection between Med cyclone strength and max jet velocity is comparable to that for Atlantic cyclones for jet velocities < 50 m/s. I think it is worth including this important nuance in this sentence.

Figure 12 caption: Please state in the caption what the dots mean on these panels.

Line 403: I recommend added the word "three"  before "seasons" for added clarity, since I assume the authors are only referring MAM, SON and DJF.

Line 404: I think this should say 4 hPa higher*, unless I am misunderstanding the sentence.

**Grammar and Style Remarks:**

Line 15: This sentence gets overly complex. I recommend starting a new sentence after "polar jet", with "We refer to this as the upstream-downstream…"
Line 31: Change "results" to "result" for subject-verb agreement
Line 50: Remove "e.g."

Line 51: Change "the North Atlantic cyclone" to "a North Atlantic cyclone"
Line 52: Add a semicolon after "Mediterranean, and replace "as by removing" with "when they removed"
Line 103: Change "reproduced" to "parameterized"
Line 193: Change "of the" to "off the"
Line 197: Remove "naturally"
Line 216: Change "intruded the" to "intruded into the"
Line 257: Change "shits" to "shifts"
Line 268: Change "cause" to "causes"
Fig. 7 caption: I recommend changing "asterisks" to "stars" in this caption, as what is plotted appear to be stars rather than an asterisks.
Line 350: Remove comma
Line 376: Add the phrase "initial perturbations" before the parentheses on this line.
Line 384: The sentence starting "In MAM," gets overly complex. I recommend simplifying by splitting it into two sentences.
Line 408: Add "the" before "ocean surface"
Line 429: Change "cycle" to "cyclone"
Line 429: Add "into" before "the Mediterranean"

---

## Author Comment (AC1)

egusphere-2023-2125

**The upstream-downstream mechanism of North Atlantic and Mediterranean cyclones in semi-idealized simulations**

by Alexander Scherrmann, Heini Wernli, and Emmanouil Flaounas

**Final author comments**

We would like to thank the anonymous reviewer and Florian Pantillon for their constructive comments, suggestions and remarks that helped us to improve the manuscript. Below are our detailed replies (in blue) to the individual comments (in black).

We would like to thank both Reviewers for carefully reading our manuscript and the fruitful comments they provided. Most comments have been incorporated.

**REVIEW 1**

**General Comments:**

The only additional analysis I would to see is a comparison of pairs of cases (selected based on Fig. 8) for which the Med cyclone intensity is similar, but the Atlantic cyclone intensity is different. Specifically, I would like to know how the max jet speed compares between the two cases, and where in the process chain linking Atlantic cyclones to Med cyclones there is a breakdown that leads to the limited sensitivity of downstream cyclone intensity to the upstream cyclone. For example, do the PV streamers or RWB events differ substantially between the two cases, and if so, does this imply that these nonlinear links in the upstream-downstream process chain are key sources of downstream variability/uncertainty? Or are the differences between the two cases strictly local in the sense that difference growth primarily occurs within the Mediterranean region? There is a bit too much speculation around Line 315 on this matter, and I feel that this comparative analysis could help make this part of the manuscript less speculative.

Thank you for this insightful suggestion. The deepest Mediterranean cyclones form in SPPT experiments, in which the suggested analysis cannot be conducted. However, please find attached the analysis of three particular non-SPPT experiments in winter with the following intensities of North Atlantic and Mediterranean cyclones:

| Simulation | NA cyclone intensity | Med cyclone intensity |
|---|---|---|
| 400 km north moderate | 972 hPa | 999.8 hPa |
| 400 km west moderate | 976.5 hPa | 994.2 hPa |
| 400 km north strong | 964.7 hPa | 995.7 hPa |

The initial conditions of these three experiments are shown in the following figure (same as Fig. 1 in the manuscript with PV and U at 300 hPa).

[Figure]

Fig.1

The northward shifted perturbations significantly accelerate the jet and result in larger PV values around the jet, even within the same amplitude of initial perturbation (Fig. 1a,d). In the 4N strong initial perturbation experiment the PV values and zonal velocity are significantly larger than in the 4W moderate one (Fig. 1 h,i).

The evolution of the atmospheric state and emergence of the large scale features, i.e., track of the North Atlantic cyclone, shape and amplitude of the ridge and morphology of the PV streamer are similar to each other when same amplitude of perturbation is applied (left and central column in Fig. 2). However, the PV streamer has lower PV values in the 4N moderate experiment (e.g. Fig. 2d) than in the 4W moderate experiment (Fig. 2h). Although the evolutions of the atmospheric state are comparable between the simulations, the tracks and especially the intensity of the Mediterranean cyclones are different. This might be due to slightly higher PV values in the 4W experiment, however this is not straightforward to deduce. Furthermore, comparing the 4W moderate to 4N strong experiment, we find that the NA cyclones look very different at days 2-4, but the PV streamers over the western Mediterranean develop very similarly. Also the Med cyclones have comparable tracks and reach to a very similar intensity (within 1 hPa min SLP). Already the RWB shows a lot of non-linearity in the sense that a stronger NA cyclone does not necessarily lead to a significantly higher amplitude of a PV streamer. Therefore,

we conclude that the amplitude of the initial perturbation is a good predictor for the intensity of the NAtl cyclone but not for the RWB and Med cyclone.

[Figure]

The abstract states that the sequence of events leading to Mediterranean cyclogenesis "rarely occur in a spatially consistent, fully repetitive pattern", but the results from the idealized simulations actually suggest that this sequence of events leads to Mediterranean cyclogenesis rather consistently. Details of the intensity and position of the Mediterranean cyclone may vary by time of year or with the intensity of the upstream perturbation, but the overall sequence of events leading the cyclogenesis in the Mediterranean seems quite robust in the simulations. It may help to clarify in the abstract this apparent lack of consistency in the proposed mechanism is

for real cyclone cases. Additional statements in the discussion about possible reasons why this pattern more reliably leads to cyclogenesis downstream in the simulations than in the real atmosphere would also be useful. For example, could the consistency of the mechanism in simulations be a result of using smoothed, climatological initial conditions?

> Thank you for suggesting to differentiate between the real cyclones and the consistent occurrence of the mechanism in the simulations in the abstract. And we agree with your hypothesis for the reason for this high consistency in the simulations as we already mentioned in l. 451-460: "However, this lack of sensitivity of the North Atlantic cyclone tracks and downstream intrusion of the PV streamer to the initial perturbation might be due to our choice of the climatological atmospheric states as initial conditions."

I am curious if the fact that this mechanism is absent during JJA implies that it is contingent upon there being a zonally extensive and continuous jet stream waveguide across the Atlantic, or if the lack of downstream Mediterranean cyclogenesis during JJA is simply a result of weaker upstream baroclinicity over the North Atlantic preventing the formation of North Atlantic cyclone seeds. If the existence of the waveguide is a necessary condition for the upstream-downstream mechanism, then I would also imagine that climate change impacts on the jet stream/waveguide will influence this mechanism. For example, a weaker jet in future climates could make everything more JJA-like, in which case Med cyclones triggered through this mechanism might become less common. I understand that this is well beyond the scope of the study, but it may be worth mentioning at the very end of the manuscript that climate change impacts on the jet stream could conceivably influence the prevalence of this mechanism in future climates.

> Thank you for the excellent suggestion. In the summer experiments, in which the NA cyclones are substantially weaker, the RW barely intrudes the Mediterranean as suggested by Flaounas et al. (2018). We indeed considered the future trends of the mechanism by performing similar experiments where we take the initial conditions from CESM2 simulations for the present-day and future climate, as part of an additional study of the first author's PhD project. The results will be published in the doctoral thesis in 2024, after the thesis has been reviewed. So far we can confirm that in the future climate, in winter, fewer experiments show cyclogenesis in the Mediterranean, although the polar jet intensifies and shifts southward over the North Atlantic. Therefore, the RW still intrudes the Mediterranean in the future climate, but does not always trigger cyclogenesis. Preliminary results suggest that this is due to an increased static stability over the Mediterranean. When a wave intrudes the Mediterranean in form of a PV streamer/cutoff, cyclogenesis does not take place in some experiments and as the far-field effect of the PV anomaly in the upper troposphere is reduced by the increased static stability. Those that form are weaker cyclones for the same reason and due to an increased climatological SLP inside the Mediterranean. We will mention the future climate experiments as a future study at the end of the manuscript.
>
> "In particular, such a study would allow the investigation of how the climate change impacts on the North Atlantic jet stream, i.e., its altered intensity and position, affect the prevalence of this upstream-downstream mechanism in future climates."

As an additional remark, we would like to note that the effect of climate change on the climatological atmospheric state of winter is much weaker than the seasonal difference between winter and summer of current climate. Therefore, the climatological state of current summer should not be regarded as a direct proxy for the overall atmospheric state at the end of this century.

Flaounas, E., Kotroni, V., Lagouvardos, K. *et al.* Heavy rainfall in Mediterranean cyclones. Part I: contribution of deep convection and warm conveyor belt. *Clim. Dyn.,* **50,** 2935–2949 (2018). https://doi.org/10.1007/s00382-017-3783-x

**Clarification Comments:**

Figure 3 caption: The caption should state what the difference between the left and right columns is.

The caption now includes the description of the panels in the right column.

Figure 3: I found the purple stars and cyclone tracks to be very difficult to see on these plots. Perhaps they could be outlined in white to enhance their visibility.

Thank you for highlighting the low visibility. The tracks and markers of the mature stage are now displayed in white and the cyclone features are now labeled as suggested by the 2nd reviewer.

Line 186: It's somewhat ambiguous what "the second one" refers to here; consider replacing "one" with "PV streamer" for added clarity

Changed as suggested.

Line 195: To what cyclone climatology are the authors referring here?

We now include a reference of cyclone intensities over the North Atlantic and Mediterranean.

Flaounas, E., Kotroni, V., Lagouvardos, K., and Flaounas, I.: CycloTRACK (v1.0) – tracking winter extratropical cyclones based on relative vorticity: sensitivity to data filtering and other relevant parameters, Geosci. Model Dev., 7, 1841–1853, https://doi.org/10.5194/gmd-7-1841-2014, 2014.

Figure 6 caption: Here the terminology is "perturbation amplitude", but in the Fig. 5 caption, it is referred to as a "wind anomaly" and in earlier captions it is "perturbation intensity." I recommend being consistent in the text and captions regarding the terminology for the strength of the perturbation to the jet.

> Thank you for highlighting this inconsistent labeling. We now consistently use the terminology of "perturbation amplitude" where applicable.

Line 305: Please state in the text the significance test and confidence level for assessing the statistical significance of this correlation.

> We are now explicit about the Pearson correlation coefficient and further state the confidence level: "statistically significant with the Pearson correlation coefficient sigma=-0.93, with a confidence level of p<1e-10".

Line 322: Here the authors state that the connection between Med cyclone intensity and the max upper-level jet velocity is "substantially weaker" than the connection between the Atlantic cyclone intensity and the max upper-level jet velocity. This seems an overly-simplistic statement that overlooks the fact (acknowledged in the previous paragraph) that the connection between Med cyclone strength and max jet velocity is comparable to that for Atlantic cyclones for jet velocities < 50 m/s. I think it is worth including this important nuance in this sentence.

> Added as suggested.

Figure 12 caption: Please state in the caption what the dots mean on these panels.

> Thank you for highlighting the missing description of the outliers. We now mention them in the caption of Fig. 12.

Line 403: I recommend added the word "three" before "seasons" for added clarity, since I assume the authors are only referring MAM, SON and DJF.

> Thank you for this suggestion on clarification. Added as suggested.

Line 404: I think this should say 4 hPa higher*, unless I am misunderstanding the sentence.

> Yes, thank you, that is indeed correct. Changed as suggested.

**Grammar and Style Remarks:**

Line 15: This sentence gets overly complex. I recommend starting a new sentence after "polar jet", with "We refer to this as the upstream-downstream…"

> Changed as suggested.

Line 31: Change "results" to "result" for subject-verb agreement

> Changed as suggested.

Line 50: Remove "e.g."

Changed as suggested.

Line 51: Change "the North Atlantic cyclone" to "a North Atlantic cyclone"

Changed as suggested.

Line 52: Add a semicolon after "Mediterranean, and replace "as by removing" with "when they removed"

Added as suggested.

Line 103: Change "reproduced" to "parameterized"

Changed as suggested.

Line 193: Change "of the" to "off the"

Changed to "at the east coast of North America".

Line 197: Remove "naturally"

Changed as suggested.

Line 216: Change "intruded the" to "intruded into the"

Changed as suggested.

Line 257: Change "shits" to "shifts"

Changed as suggested.

Line 268: Change "cause" to "causes"

Changed as suggested.

Fig. 7 caption: I recommend changing "asterisks" to "stars" in this caption, as what is plotted appear to be stars rather than an asterisks.

Thank you for this remark, we changed it as suggested.

Line 350: Remove comma

Changed as suggested.

Line 376: Add the phrase "initial perturbations" before the parentheses on this line.

Added as suggested.

Line 384: The sentence starting "In MAM," gets overly complex. I recommend simplifying by splitting it into two sentences.

We now simplified the sentence as follows:

"In spring cyclogenesis takes places over Algeria for the weak and moderate initial perturbations. Afterwards, these cyclones move to the east and west of Italy, respectively (Fig. 11d,e). In the strong perturbation experiments cyclones form and remain over the western Mediterranean Sea, i.e., the Balearic Sea and Gulf of Genoa."

Line 408: Add "the" before "ocean surface"

Added as suggested.

Line 429: Change "cycle" to "cyclone"

Changed as suggested.

Line 429: Add "into" before "the Mediterranean"

Added as suggested.

**General comments**

The paper seems to claim that Atlantic cyclones are the cause of Mediterranean cyclones. However, in the used numerical framework the initial perturbation is applied to upper-levels dynamics, which contributes to Atlantic cyclogenesis but also triggers Rossby waves that propagate further downstream, thus the link between Atlantic and Mediterranean cyclones may not be causal. Hovmöller plots of flow anomalies compared to the background climatological state may help investigating the respective contributions of Atlantic cyclogenesis and Rossby wave propagation.

> We fully agree that we are too absolute in our formulation, although we mean to say the same. Making reference to "Atlantic cyclone causing Mediterranean cyclogenesis" we mean that the former modulates the (downstream) evolution of the atmosphere by amplifying the ridge that leads to RWB, the intrusion of the PV streamer and thereby to Mediterranean cyclogenesis. We are now more careful with our formulation of causality. As suggested, please find below the averaged over all non-SPPT experiments in winter (which all show the same pattern with different amplitude) Hovmöller analysis of the v-wind component at 300 hPa in the depicted domain (Fig. 3). It confirms the propagation of the RW induced by the initial perturbation and the intrusion of the PV streamer over the Mediterranean after about 4d.

[Figure]

Fig. 3

The abstract and introduction emphasize the role of warm conveyor belts in the chain of events that leads to Mediterranean cyclogenesis but these are not discussed further in the paper (although they are implicitly accounted for via the model microphysics). Quantifying their contribution would likely need new simulations with the microphysics scheme modified or switched off, which may be too demanding, but some estimate or investigation would be appreciated.

To quantify the effects of the WCBs, we performed experiments with a weak, moderate and strong initial perturbation positioned at the maximum zonal jet at 300 hPa with the microphysics and convection parameterization switched off. The Atlantic cyclone is significantly weaker compared to the experiments with the full use of parameterization. In all experiments we observe the eastward propagation of the RW induced by the initial perturbation. However, we do not observe RWB and thereby the intrusion of a PV streamer into the Mediterranean. Nonetheless, as the RW propagates eastwards, a trough intrudes the Mediterranean (see Figure below for the moderate initial perturbation after 2d (a) and 4d (b)). The PV values of the trough are lower by 2-3 PVU compared to the experiments with microphysics and convection being activated (compare Fig. 4b to Fig.

3h in the manuscript). The trough does not lead to cyclogenesis in the Mediterranean in the experiments with a weak or moderate initial perturbation. In the case of the strong perturbation experiment, the cyclone tracking algorithm identifies a Mediterranean cyclone with a central pressure of about 1015 hPa after 3.5 d, west of Italy. However, it is debatable whether this cyclone corresponds to an actual well organized meso-scale vortex. Furthermore, given that in the experiment with a strong initial perturbation the anomaly exceeds twice the seasonal variability (comment on seasonal variability below) of the wind speed at 300 hPa of the atmosphere, the induced RW might already show a strong oscillation and hence southward intrusion with significantly large PV values, leading to the onset of cyclogenesis. Therefore, we conclude that the diabatic processes within the WCBs (and thereby the WCBs as shown by multiple studies) of the North Atlantic cyclone are an essential element in provoking RWB, for the consequent intrusion of the PV streamer over the Mediterranean and eventually for Mediterranean cyclogenesis.

[Figure]

The results do not really reflect the actual seasonal and geographical distribution of Mediterranean cyclones, especially the hotspot over the Gulf of Genoa. This questions how the location and intensity of PV/jet perturbations in the semi-idealized framework compare with the observed variability around the climatological composites. Again, re-running the simulations would likely be too demanding but the perturbations should at least be compared to the actual variability for each season.

Although we take the climatological mean atmospheric state for the initial conditions, we should not expect the evolution of the average flow to represent the mean of variously different daily atmospheric states. In real atmospheric conditions, the non-linearity of the processes involved result in a wide variety of different evolutions than the ones met in our semi-idealized setup with a single climatological mean state. We cannot resample the full variability of the atmosphere and thus cannot reproduce the full spectrum of morphologies, locations and amplitudes of atmospheric features involved (North Atlantic cyclone, ridge, PV streamer, Mediterranean cyclones, and also everything we do not explicitly address in our analysis).

Nonetheless, comparing our perturbations to the actual variability of the atmosphere is a valuable suggestion. We calculated the standard deviation (std) of PV and wind speed at 300 hPa with respect to the seasonal climatologies in the same time period. The following table lists the standard deviation over the same time period from ERA5, respect to the amplitude of the anomaly due to the presence of the initial perturbations in our experiments. Calculations have been done at 300 hPa averaged within a box of 500 km around the maximum wind speed of the climatological jet at 300 hPa. Tests with boxes of 250-1000 km yielded comparable values.

| Season | std wind@300 hPa | std PV@300 hPa | PV anomaly of a weak (11 m s$^{-1}$) perturbation | PV anomaly of a moderate (22 m s$^{-1}$) perturbation | PV anomaly of a strong (34 m s$^{-1}$) perturbation |
|---|---|---|---|---|---|
| Winter | 16.4 m s$^{-1}$ | 1.8 PVU | 1.5 PVU | 3.7 PVU | 6.5 PVU |
| Spring | 15.4 m s$^{-1}$ | 1.63 PVU | 1.1 PVU | 2.7 PVU | 4.7 PVU |
| Summer | 13 m s$^{-1}$ | 1.65 PVU | 1.2 PVU | 2.9 PVU | 5 PVU |
| Autumn | 15.8 m s$^{-1}$ | 2.1 PVU | 1.5 PVU | 3.6 PVU | 6.2 PVU |

The PV values of the (strong) perturbations exceed those of the seasonal variability, however, these represent (when added to the climatological state) realistic PV values of very intense troughs. The maximum wind speed values of our perturbations are of the order of the seasonal variability of the wind speed at 300 hPa. Thus, our perturbations range from 0.85 to 2.6 times the seasonal variability and thus are a reasonable choice, given the daily variability of the atmosphere. Note that additional experiments with weaker perturbations were performed (5 & 8 m s$^{-1}$ that correspond to 0.3 to 0.61 the seasonal variability, included in Table 1 of the first version of the manuscript), but did not provide further insights into the connection and its sensitivity.

The added value of running perturbed experiments with the SPPT scheme is unclear, because the results are not discussed much. Perhaps this would contribute answering comment 2 on the role of warm conveyor belts, as suggested in Section 2.2.

The motivation for the SPPT experiments was that a plethora of processes, or atmospheric

systems, are involved into the upstream-downstream connection. RWB is the centerpiece of the connection and is directly relevant to large scale atmospheric circulation. However, the morphology of the RWB is also a factor of diabatic processes and momentum forcings from turbulence (Gray, 2006; Spreitzer et al., 2019). In addition, the North Atlantic cyclone intensity and WCBs are also expected to interact with the RW and modulate accordingly its propagation and breaking characteristics. In fact, it was the hypothesis of Raveh-Rubin and Flaounas (2017) that the intensity of North Atlantic cyclones correlates with the one of Mediterranean cyclones. Consequently, diabatic processes potentially play a crucial role in the upstream-downstream connection. In a five-day model integration and within such a large geographical domain, it is rather difficult to pinpoint the exact atmospheric processes or locations where the upstream-downstream connection is sensitive to minor momentum or temperature diabatic perturbations. In these regards, the ensemble of SPPT experiments provide adequate means to investigate the robustness of the connection and to reply to the question whether we should expect "big" changes in the location and intensity of Mediterranean cyclones due to "small" upstream changes, over the Atlantic Ocean.

However, we agree that the SPPT experiments do not significantly contribute to the key results of the paper and thus we remove them as suggested. Please find attached (Fig. 5-7) the new versions of Fig. 8, 11 and 12 of the manuscript. We adjust the manuscript in the corresponding lines.

Gray, S. L. (2006), Mechanisms of midlatitude cross-tropopause transport using a potential vorticity budget approach, *J. Geophys. Res.*, 111, D17113, doi:10.1029/2005JD006259.

Spreitzer, E., R. Attinger, M. Boettcher, R. Forbes, H. Wernli, and H. Joos, 2019: Modification of Potential Vorticity near the Tropopause by Nonconservative Processes in the ECMWF Model. *J. Atmos. Sci.*, **76**, 1709–1726, https://doi.org/10.1175/JAS-D-18-0295.1.

[Figure]

Fig 5.

[Figure]

Fig 6.

[Figure]

Fig. 7

I recommend writing spring, summer, autumn and winter in the text instead of MAM, JJA, SON and DJF but I leave it to the authors' preference.

Changed as suggested.

**Specific comments**

Title: "upstream-downstream mechanism of cyclones" sounds awkward; for instance, l. 15–16 rather define the "upstream-downstream mechanism of cyclogenesis" and l. 69–70 refer to an "upstream-downstream connection of cyclones"

We now consistently use the wording of "upstream-downstream connection (of cyclones)".

The first third of the abstract solely describes results from previous studies and with specific information, which appears unnecessary, while more details are expected on the actual results of

the paper described in the last third of the abstract. For instance, what is the sensitivity of the Mediterranean cyclone characteristics to the dynamical structure and intensity of the intruding PV streamer? How does the seasonal cycle of Mediterranean cyclogenesis depend on the large-scale atmospheric circulation?

> We adjusted the abstract accordingly and now include more quantitative results.

l. 29–30 Which are "The spatially distinct regions of cyclogenesis in the Mediterranean"?

> We mean that PV streamers over the western and eastern parts of the Mediterranean lead to cyclogenesis in the corresponding region and thereby create cyclones in "distinct" regions. We now clarify that in the text.

l. 35–36 Not sure what is meant by "south of the Alps" for Argence et al. (2008): in their abstract, "This study explores the predictability of a heavy rainfall event that struck North Africa on 9 and 10 November 2001"

> We are now more explicit with our formulation regarding the emergence of heavy precipitation in the western parts of the Mediterranean and explicitly differentiate between these two events over the Gulf of Genoa and North Africa.

l. 51 A tropical cyclone is involved in this specific case

> We are now more explicit with "a tropical cyclone over the North Atlantic, that underwent extratropical transition".

l. 112 Repetition of l. 102

> We removed the repetition.

l. 125 Repetition of l. 115

> We removed the repetition.

l. 131 Remove "both" here

> Removed as suggested.

l. 135 northeastward?

> Thank you for highlighting the missing and the more accurate east part.

l. 137 upper troposphere

> Changed as suggested.

l. 146 x, y and z − zh should be |x-xh|, |y-yh| and |z − zh| in Equation 1

> Thank you for highlighting the missing differences and absolute values. Added as suggested.

Table 1 The added value of indicating "simulations that are not explicitly discussed as they provided no additional insight" appears limited

> We agree that this does not provide further insights for the results and therefore we removed them and shortened Table 1 accordingly.

l. 171–175 A brief description is expected here for the method of Wernli and Schwierz (2006), and for the modifications of Sprenger et al. (2017) if relevant

> We now include a short description of the methodology.

l. 178–194 Identifying the relevant features on Figure 3 is not obvious; labeling them would help

> Thank you for that suggestion. In addition to changing the colors to white, as suggested by Reviewer 1, we further include a labeling of the North Atlantic and Mediterranean cyclones.

l. 236 The terminology should be introduced in Section 2 and included in Table 1

> Now included in Table 1 as suggested.

l. 260–261 This contradicts the above statement that "similar results are found for the strong and weak initial perturbation"

> We find a similarly low spreads of the PV streamers for the strong and weak initial perturbation as for the moderate ones. However, when changing the amplitude of the perturbation we get a more pronounced response compared to when changing the horizontal position. We now articulate better to avoid confusion.

l. 273–274 This is already described in Section 2.2 (but with 10 ensemble simulations)

> This is a misunderstanding. Figure 6 does not include the 10 SPPT experiments but shows the composite of the 9 basic horizontal shifted experiments, i.e., 1 at the center, 4 shifted by 200 and 4 by 400 km, respectively. We are now more explicit in the text to avoid confusion.

> However, a further analysis shows that the spread of PV streamers between the SPPT experiments with the same initial conditions is about 1-4 degree in longitude and comparable to horizontal shifts in Figure 4.

l. 276 How large is the spread between simulations?

> We now quantify the spread between the simulations in the text. The spread of the 2 PVU contour at 300 hPa (relative to the center experiments) is up to 2.5° in longitude, 4° in

longitude, and 5.5° in longitude for the weak, moderate and strong perturbation experiments, respectively.

l. 282 Where?

We now include the reference to Sect. 5.2.

l. 305 Is sigma the correlation coefficient?

Yes, it is. We are now more explicit in the text: "statistically significant with the Pearson correlation coefficient sigma=-0.93, with a confidence level of p<1e-10."

l. 306–307 Please explicit the dynamical link between jet streak intensity and large-scale forcing for upward motion that amplifies the development of the cyclone.

Here we apply basic quasi-geostrophic concepts. A typical Q-vector pattern of a jet streak shows Q-vector convergence and hence large-scale forcing for ascent in the left exit of a jet streak. And the intensity of the Q-vector is proportional to the baroclinicity and therefore to the intensity of the jet.

l. 322-323 Is the "heavy influence of local factors like sea surface temperature and orography in the Mediterranean" shown somewhere, or is it an assumption based on previous studies?

It is an assumption based on previous studies and we now include explicit references.

l. 387 Refer to Fig. 11f

Added as suggested.

l. 390 Repetition of l. 321–322

We believe that the reviewer is referring to l. 221-222 instead of 321-322, as in 321-322 we speak of North Atlantic cyclones whereas in l. 390 we speak of the consistency of Mediterranean cyclone tracks with the results of Almazroui et al., which we also do in lines 221-222. We remove the statement in lines 221-222 and keep it in 390, as there we discuss the tracks of all winter experiments in l. 390.

l. 392 What are the cyclogenesis hotspots found by Bartholy et al. (2009)?

The hotspots (their Fig. 4) are the Gulf of Genoa, the Balearic Sea, the west coast of Spain. We now explicitly mention them in the text.

l. 401 This contradicts the discussion in l. 318–317

We now correct the text in l. 314-317 and are more careful on the description of the figure. In winter the amplitudes are comparable for moderate and strong initial perturbations (red). However, in our simulations the Mediterranean cyclone's intensity

increases when comparing weak to moderate perturbation experiments. In autumn the intensity also increases from moderate to strong initial perturbation experiments.

"In all experiments, the average SLP (red) shows a stronger decrease with increasing amplitude of the initial perturbation (columns in Fig. 12). Here, higher-amplitude PV streamers provide a stronger baroclinic forcing and thus a stronger intensification of the Mediterranean cyclones. Further, when comparing the weak to moderate amplitude experiments (left and central column in Fig. 12), the cyclones reach lower minimum SLP values and thereby a larger maximum intensity. The intensity of Mediterranean cyclones in the strong perturbation experiments (right column in Fig. 12) is similar in winter (Fig. 12c), weaker in spring (Fig. 12f) and stronger in autumn (Fig. 12i) compared to the moderate perturbation"

l. 404 higher

Changed as suggested.

l. 411-412 Any reference for that? I would expect the strongest cyclones to develop in winter

This statement was based on a climatological analysis of 3000 cyclones in Scherrmann et al. (2023), however we made a mistake and now find equal average intensities of cyclones in winter and spring. We removed the statement. The deeper cyclones in our experiments are probably due to the explanation in l.412-414.

l. 412–414 SSTs are lowest in spring

Given the difference of our initial climatological average, SSTs in the western and central parts of the Med are higher in spring than in winter. Higher SSTs have been shown in (please note the changing colorbar scale):

Shaltout, M., & Omstedt, A. (2014). Recent sea surface temperature trends and future scenarios for the Mediterranean Sea. Oceanologia, 56(3), 411-443. https://doi.org/10.5697/oc.56-3.411

We now refer to their study for reference.

l. 418–419 This appears to contradict l. 412–414

We correct our statement. We are still convinced of the statement in l.412-414 for the weak and moderate initial perturbation experiments. We further tested our hypothesis of l.418-419 but could not find a systematic pattern of PV erosion at upper levels (although in some cases this seems to occur). We do not understand why in the strong perturbation experiments the intensity is comparably low and clarify this now in the text. Our assumption of PV erosion was based on the findings of Flaounas et al. (2021) but potentially, very strong convection displaces the PV streamer and thus reduces its effect

on the surface cyclone. However, there is no clear evidence for our suggestion, indicating that further analysis is required to fully understand this behavior. The text now reads:

"We cannot explain why high-amplitude PV streamers in experiments with strong initial perturbations do not further increase the intensity of the Mediterranean cyclones. Considering the results of Flaounas et al. (2021), who showed that the stronger the upper-level forcing of Mediterranean cyclones the weaker is the diabatic forcing, there might be a balancing factor that makes it difficult to attribute a straightforward connection between the amplitude of the PV streamers and the intensity of the cyclones. We have two hypotheses where one is that convection is exceptionally stronger in the strong perturbation experiments compared to the moderate ones and thus might erode the upper-level PV and thereby inhibit its maximum contribution to the intensity of the cyclones. The second hypothesis involves the displacement of the PV streamer by low-PV air that is transported to the upper-troposphere and thereby results in a less favorable alignment of the PV streamer and the cyclone, yielding a reduced intensity. However, both hypotheses require further investigation, as we could only find some cases that agree with one of the hypotheses but no systematic pattern."

l. 433 "the the"

Removed as suggested.

l. 475 Why is the static stability reduced in MAM? SSTs are lowest in spring (see l. 412–414)

We conclude about the reduced static stability in spring due to a reduced vertical gradient of potential temperature, i.e., the following quantity

$$[\theta_{MAM}(300hPa) - \theta_{MAM}(900hPa)] - [\theta_{DJF}(300hPa) - \theta_{DJF}(900hPa)]$$

is typically negative with values of -2 K over the western and up to -4 K over the eastern parts of the Mediterranean (Fig. 8).

[Figure]

Fig. 8

l. 481 Not exactly: the Atlantic cyclogenesis is triggered by the upper-level PV anomaly, which also impacts Rossby wave propagation, thus there may be no strictly causal link between the Atlantic and Mediterranean cyclones

We agree that this covers our implicit causality of the connection and that we are too absolute with the formulation. We adjusted the line accordingly and it now reads:

"In our study, we investigated how North Atlantic cyclones, triggered by an upper-level PV anomaly, affect the propagation of the RW and the RW's impact on downstream cyclone dynamics over the Mediterranean."

---

## Author Response (AR2)

egusphere-2023-2125

**The upstream-downstream mechanism of North Atlantic and Mediterranean cyclones in semi-idealized simulations**

by Alexander Scherrmann, Heini Wernli, and Emmanouil Flaounas

**Final author comments**

We would like to thank the anonymous reviewer and Florian Pantillon for their constructive comments, suggestions and remarks that helped us to improve the manuscript. Below are our detailed replies (in blue) to the individual comments (in black).

We would like to thank both Reviewers for carefully reading our manuscript and the fruitful comments they provided. Most comments have been incorporated.

**REVIEW 2**

The authors satisfactorily addressed my previous main concerns: they clarified the nature of the connection of North Atlantic and Mediterranean cyclones, performed experiments with the microphysics and convection parameterization switched off to quantify the role of warm conveyor belts, compared the amplitude of the perturbations to the actual variability for each season, and removed unnecessary mentions to experiments with SPPT perturbations.

However, while I appreciate the effort, I am surprised that the additional work described in the response to the reviewers is not mentioned in the revision, especially the performed experiments concerning the role of WCBs. I understand the authors may not want to extend the paper with an additional figure but I expect the results to be mentioned, as WCBs are a key ingredient in the chain of events that leads to Mediterranean cyclogenesis and as such are referred to in the abstract.

> We now include the dry-sensitivity experiments in the Appendix

Minor comment
I think there is a confusion between trends and absolute values of SST when referring to Shaltout and Omstedt (2014). The Mediterranean SST is definitely lower in spring than in autumn, e.g., see http://www.ceam.es/ceamet/SST/SST-climatology.html

> Indeed, there was a misunderstanding. We now removed the reference and corrected our statement in the text.

Typos
l. 383 and and

> Removed as suggested.

l. 422 the weaker the diabatic forcing (no is)

Removed as suggested.

l. 448 This suggests that (no comma)

Removed as suggested.